# Cortical astrocyte histamine-1-receptors regulate intracellular calcium and extracellular adenosine dynamics across sleep and wake

Charlotte R. Taylor[1,2], Vincent Tse[1☯], Drew D. Willoughby[1,2☯], Maxine Levesque[1,2], Trisha V. Vaidyanathan[1,2], Jeanne T. Paz[2,3,4], Kira E. Poskanzer[1,2,4*]

1 Department of Biochemistry & Biophysics, University of California, San Francisco, San Francisco, California, United States of America, 2 Neuroscience Graduate Program, University of California, San Francisco, San Francisco, California, United States of America, 3 Gladstone Institute of Neurological Disease, University of California, San Francisco, San Francisco, California, United States of America, 4 Department of Neurology, and Kavli Institute for Fundamental Neuroscience, University of California, San Francisco, San Francisco, California, United States of America

☯ These authors contributed equally to this work.
* kira.poskanzer@ucsf.edu

## Abstract

Classical neuromodulators regulate arousal states, spanning deep sleep to vigilant wakefulness, primarily by activating cortical neurons. However, cortical astrocytes also express neuromodulatory G-protein-coupled receptors (GPCRs). While astrocytic noradrenergic receptors have been shown to modulate two critical regulators of arousal—cortical synchrony and extracellular adenosine levels—how other neuromodulatory signaling pathways similarly shape arousal remains unclear. Astrocytes in mammalian cortex express particularly high levels of the wake-promoting, Gq-coupled histamine-1-receptor (H1R), yet little is known about how astrocytic H1R contributes to regulation of arousal. To address this gap, we used pharmacological and genetic approaches in murine cortex to test how astrocyte-H1R signaling affects astrocyte calcium ($Ca^{2+}$), cortical neural activity across sleep/wake, and extracellular adenosine—an astrocytic output that regulates cortical arousal. Using ex vivo two-photon $Ca^{2+}$ imaging in acute cortical slices, we show that H1R mediates cell-autonomous astrocyte $Ca^{2+}$ responses to histamine (HA) and attenuates responses to norepinephrine (NE). Next, in vivo fiber photometry and electrophysiology results show that H1R deletion in cortical astrocytes disrupts local astrocyte $Ca^{2+}$ during wake and extracellular adenosine dynamics specifically around REM sleep transitions, when HA release is minimal. Further, astrocyte-specific H1R deletion in cortex promotes wakefulness and reduces REM sleep time. Our results indicate that H1R activity modulates astrocyte responses to non-histaminergic inputs by inducing lasting changes in astrocyte physiology that modulate extracellular adenosine and REM sleep. Our findings contribute to an emerging model in which neuromodulator

**Data availability statement:** All data underlying the results are available in Dryad (https://doi.org/10.5061/dryad.2280gb64x). The analysis scripts and notebooks used to generate the figures are archived in Zenodo (https://doi.org/10.5281/zenodo.16809849). Both resources are publicly available without restriction.

**Funding:** This work was supported by funding from the National Institutes of Health (https://www.nih.gov/; project grants R01NS099254 and R01MH121446) and the National Science Foundation (https://www.nsf.gov/; NSF CAREER 1942360) to KP. The funders played no role in study design, data collection and analysis, decision to publish, or preparation of the manuscript.

**Competing interests:** The authors have declared that no competing interests exist.

**Abbreviations:** ACh, acetylcholine; ACSF, artificial cerebrospinal fluid; AUC, area under the curve; cKO, conditional knockout; DA, dopamine; EEG, electroencephalogram; FOV, field-of-view; GPCRs, G-protein-coupled receptors; H1R, histamine-1-receptor; HA, histamine; HMM, Hidden Markov model; IACUC, Institutional Animal Care and Use Committee; IHC, immunohistochemistry; NE, norepinephrine; PSD, power spectral density; ROIs, regions of interest; RT, room temperature; V1, visual cortex; WT, wild-type.

GPCRs synergistically shape astrocyte physiology to regulate arousal behavior and adenosine signaling in the cortex.

## Introduction

The balance of sleep and wake is fundamental to animal survival. These distinct behavioral states are orchestrated in part by subcortical nuclei that project to and regulate the cortex via release of different neuromodulators, including norepinephrine (NE), acetylcholine (ACh), dopamine (DA), serotonin, and histamine (HA) [1–3]. These neuromodulators promote wakefulness by binding neuromodulatory G-protein-coupled receptors (GPCRs) on cortical neurons, which increases cortical desynchrony, an indicator of arousal [1–3]. However, neurons are not the only cortical cell type expressing these receptors; several subtypes of cortical glia, including astrocytes, microglia, and oligodendrocyte precursor cells, also express neuromodulatory GPCRs [4–6] and respond to their activation [7–15]. Yet, little is known about how glial integration of neuromodulators contributes to arousal control.

Astrocytes—the most numerous glial cell type in the brain—have been shown to regulate cortical activity underlying sleep/wake through intracellular calcium ($Ca^{2+}$) signaling [16,17]. Cortical astrocytes sense changes in sleep/wake via $Ca^{2+}$, exhibiting a clear positive correlation between intracellular $Ca^{2+}$ levels and increased arousal [18–24]. Additionally, stimulation of astrocyte $Ca^{2+}$ promotes cortical oscillations in the delta frequency range (0.5–4 Hz) that define non-rapid eye movement (NREM) sleep [24–26]. Astrocyte-mediated changes in cortical neuronal activity can occur across many minutes after stimulation of astrocyte $Ca^{2+}$ during wakefulness [27] leading to delta power changes during NREM sleep periods [24]. Astrocytic $Ca^{2+}$ has been shown to promote delta oscillations via adenosine release [28,29], which accumulates extracellularly across wakefulness [30,31] and can increase local delta power in the cortex [32,33]. Deletion of astrocytic adenosine kinase also triggers both increased extracellular adenosine and NREM delta power [34], reinforcing the link between astrocyte activity and adenosine-mediated changes in the cortical synchrony of NREM. Conversely, attenuation of astrocyte $Ca^{2+}$ signaling throughout the brain can reduce delta power [18,19], and increase both theta power and time spent in REM [35]. Together, these studies suggest that astrocytes may integrate arousal levels to homeostatically promote sleep or reduce arousal via GPCR-mediated signaling in response to broad changes in neuromodulatory tone.

Our understanding of how neuromodulators contribute to astrocytic arousal control is almost exclusively informed by noradrenergic studies. Several studies have shown that NE couples astrocyte $Ca^{2+}$ to arousal events within wake periods [21–23]. In addition, astrocytic alpha-1a adrenergic receptors (Adra1a) resynchronize cortical dynamics within seconds following NE-induced desynchronization during wakefulness [36], and astrocytic Adra1a receptors can regulate neuronal synchrony by increasing extracellular adenosine [37–39]. These studies support a model in which astrocytes respond to NE via $Ca^{2+}$ signaling to regulate key indicators of reduced arousal: increased cortical synchrony [36] and adenosine release [38–41]. However,

astrocytes express many other neuromodulatory GPCRs [4,5], and it remains unclear how non-adrenergic neuromodulators are integrated by astrocytes to contribute to arousal regulation, an important gap to address since distinct neuromodulators can exert synergistic effects on astrocyte activity [42].

RNA-sequencing data [5,6] show that astrocytes express high levels of the wake-promoting histamine-1-receptor (H1R) [43–53] relative to other neuromodulatory GPCRs, but how H1R signaling contributes to arousal regulation via astrocytes is under-explored. Previous work demonstrated that brain-wide, mosaic astrocyte-specific H1R deletion led to reduced arousal during wake periods [54]. However, the role of astrocytic H1R in specific brain areas, H1R's impact on astrocyte physiology, and the mechanisms underlying astrocytic H1R-dependent changes in arousal remain unclear. Here, we investigated how HA modulates astrocyte $Ca^{2+}$, and tested whether astrocyte-HA signaling in cortex regulates extracellular adenosine levels and cortical dynamics, two readouts of arousal.

Based on evidence that astrocyte $Ca^{2+}$ signaling in visual cortex (V1) can shape arousal [24,36], we used a V1-targeted viral conditional knockout (cKO) strategy in mice and two-photon (2P) $Ca^{2+}$ imaging in cortical slices to characterize H1R signaling in astrocytes. We show that astrocyte-specific H1R mediates HA-triggered astrocyte $Ca^{2+}$ elevations and that H1R activity attenuates astrocyte $Ca^{2+}$ responses to NE. Using in vivo fiber photometry and electrophysiology in wild-type (WT) and H1R cKO mice, we demonstrate that astrocytic H1R deletion counterintuitively increases astrocyte $Ca^{2+}$ activity during wake, suggesting that H1R activity modulates how astrocytes respond to non-histaminergic inputs across sleep/wake, consistent with H1R-mediated changes in astrocytic responses to NE. Next, using in vivo fiber photometry recordings of extracellular adenosine [55] and electrophysiology, we show that astrocytic H1R deletion reduces extracellular adenosine levels during REM sleep, but has no effect on cortical oscillatory dynamics as measured via electroencephalogram (EEG) recordings. Finally, contrary to our initial hypothesis, cortical astrocyte-specific deletion of H1R has a wake-promoting effect and reduces REM sleep time when measured during the light phase of the animals' circadian rhythm.

Our results demonstrate that HA directly activates astrocytes via H1R, which modulates wake-specific $Ca^{2+}$ dynamics and REM-specific adenosine dynamics in the cortex with concomitant changes in wakefulness and REM sleep dynamics. Since we observe counterintuitive physiological effects of H1R deletion during periods when HA is not released and demonstrate that H1R activity modulates astrocytic responses to NE, our findings suggest that H1R regulates astrocyte responses to non-histaminergic inputs across sleep/wake. These findings strengthen the emerging view that astrocytes integrate neuromodulators to regulate arousal via extracellular adenosine, and underscore the need to clarify how neuromodulatory GPCR pathways act synergistically within astrocytes to govern purinergic control of arousal and circuit dynamics.

## Results

### HA triggers dose-dependent $Ca^{2+}$ responses in cortical astrocytes

RNA-sequencing data show that cortical astrocytes express high levels of HA receptors relative to other neuromodulatory GPCRs [5,6]. However, how HA modulates $Ca^{2+}$ in these cells remains unclear. We therefore tested whether HA drives dose-dependent $Ca^{2+}$ elevations in cortical astrocytes using 2P imaging to record an astrocytic $Ca^{2+}$ sensor (GCaMP6f) in adult, acute V1 slices before and after bath-applying a range of HA concentrations. We tested 0.5, 5, and 100 µM HA based on published data showing Bergmann glia respond to 1–100 µM HA with an $EC_{50}$ of 10 µM [11]. Since neurons also express HA receptors, we applied 1 µM TTX in the bath to block neuronal action potentials (Fig 1A). We quantified astrocyte $Ca^{2+}$ activity using the event-detection software AQuA [56] (Fig 1B).

To capture changes in both $Ca^{2+}$ event rate and area (Figs 1C, S1A, and S1B), we quantified the change in percent active pixels before and after HA addition (Δ pixels active, Fig 1D) [57] and the area under the Δ pixels active curve (AUC, Fig 1D and 1E). We found that all HA concentrations triggered an increase in Δ pixels active compared to baseline (Fig 1D) and observed dose-dependent relationships for both maximum Δ pixels active and AUC (Fig 1E). Concentration-dependent differences in these two features (Fig 1E) highlight the different temporal response dynamics triggered by each HA concentration. 0.5 µM HA triggered temporally discrete $Ca^{2+}$ events, 5 µM triggered a more

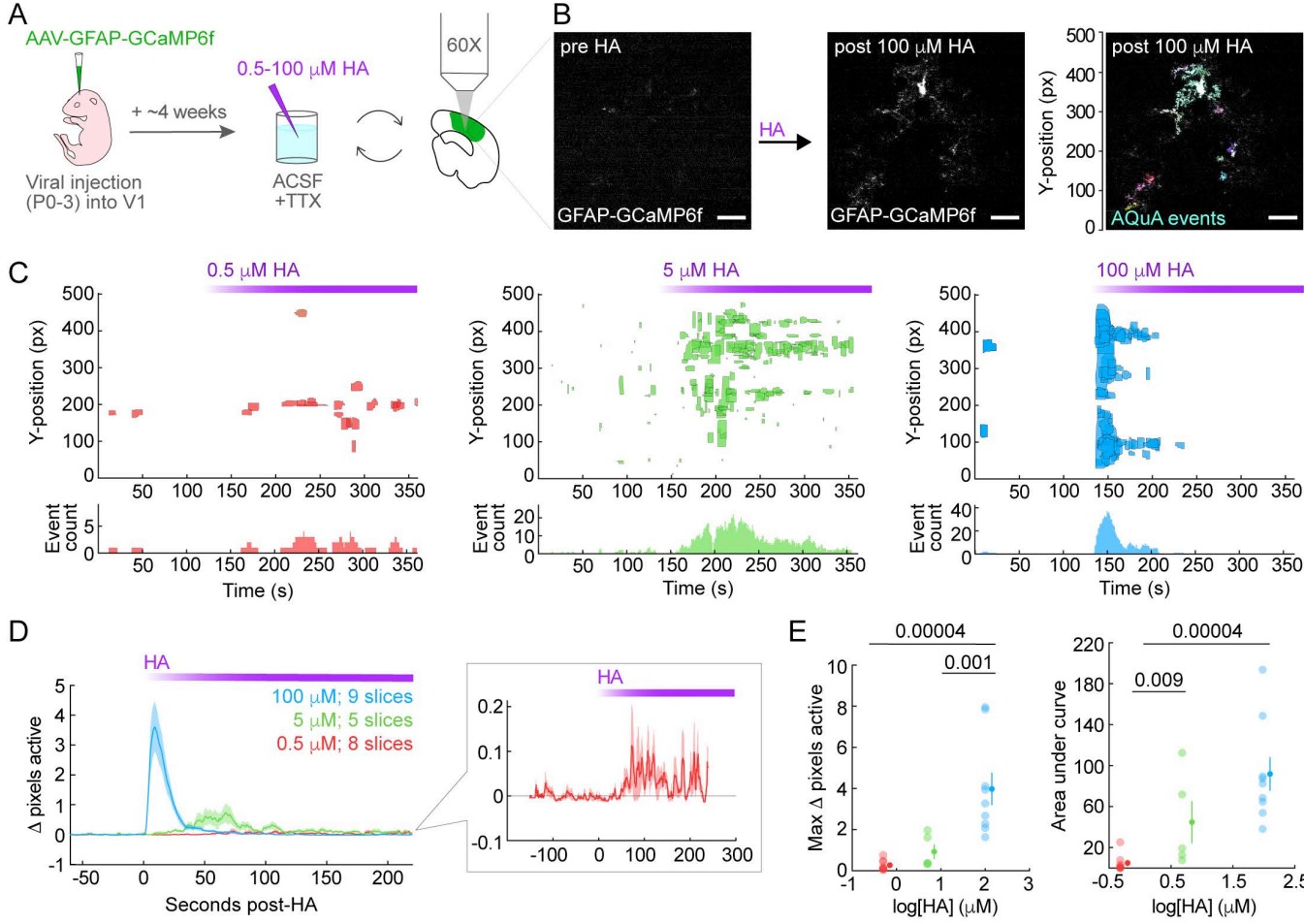

**Fig 1. Histamine (HA) drives dose-dependent Ca²⁺ responses in cortical astrocytes. (A)** Experimental schematic. Neonatal virus injection induces GCaMP6f expression in cortical astrocytes. After >4 weeks, acute V1 slices are 2P-imaged before and after bath-application of 0.5, 5, or 100 μM HA. 1 μM TTX is in bath to block action potentials for the entirety of experiments. **(B)** Four summed frames of 2P time-series show representative GCaMP6f fluorescence before (left) and after (middle) 100 μM HA addition. Right: GCaMP6f fluorescence post-HA with overlaid AQuA-detected events and y-position within FOV. Scale bars = 25 μm. **(C)** Y-axis position and duration of Ca²⁺ events before and after addition of 0.5 (left, red), 5 (middle, green), or 100 (right, blue) μM HA. Lower bar plots show number of active events per second. Purple bar = HA recirculating in ACSF, for C and D. **(D)** Change in % active pixels relative to mean 60 s pre-HA. Traces = mean across slices. Shaded error bars = standard error of mean (SEM). Inset = activity triggered by 0.5 μM HA. **(E)** Summary of data in D. Left: maximum Δ pixels active for each slice during 2-min post-HA, with mean ± SEM at right (0.5 μM: 0.3 ± 0.1, 5 μM: 0.9 ± 0.4, 100 μM: 4.0 ± 0.8). Right: area under the curve (AUC) for each slice during 2-min post-HA, with mean ± SEM at right (0.5 μM: 4.8 ± 3.0, 5 μM: 44.7 ± 20.5, 100 μM: 91.8 ± 16.4). *p*-values calculated via one-sided Wilcoxon rank-sum test. Underlying data available on Dryad (https://doi.org/10.5061/dryad.2280gb64x); panels B–E: Fig1.zip, Fig1_DoseResponse_data.mat.

sustained increase in Ca²⁺ event number, and 100 μM HA triggered a highly synchronized, relatively brief increase in event number with a concomitant increase in event size and duration (Figs 1C, 1D, and S1; S1 and S2 Videos). The amplitude, area, and duration of discrete events also changed in a concentration-dependent manner (S1C Fig). 0.5 μM HA did not affect amplitude or duration and only triggered a small increase in event area, while 5 and 100 μM clearly increased all three event metrics (S1C Fig). The spatiotemporally discrete astrocyte Ca²⁺ dynamics observed here in response to 0.5 and 5 μM HA likely reflect physiological astrocyte Ca²⁺ responses to HA, since microdialysis [58,59] and fast-scanning voltammetry data [60,61] suggest that in vivo extracellular [HA] is between 1.25 nM and 8 μM. Overall, these

results demonstrate that cortical astrocytes respond to a wide range of HA concentrations that likely falls within the physiological range of extracellular HA in cortex.

## HA-triggered astrocyte Ca²⁺ activity is H1R-dependent

Because RNA-sequencing data show that cortical astrocytes express higher levels of H1R relative to other histamine receptors (H2R, H3R) [5], we hypothesized that the HA responses we observed would require H1R activation, consistent with Gq-coupling of H1R and with astrocyte Ca²⁺ data from cerebellum [11], hippocampus [12], and olfactory bulb [13]. To test whether H1R mediates HA-triggered cortical astrocyte Ca²⁺, we 2P-imaged astrocyte GCaMP6f activity while pharmacologically blocking H1R (Fig 2A, 50 µM chlorpheniramine), and bath-applied 50 µM HA, a concentration that triggers robust Ca²⁺ elevations based on our dose–response data (Fig 1). Inhibition of H1R completely abolished the HA-triggered astrocyte Ca²⁺ response (Fig 2B–2D), including HA-triggered increases in Ca²⁺ event amplitude, duration, and area (Fig 2E), suggesting that HA-triggered astrocyte Ca²⁺ is mediated by H1R with little-to-no contribution by H2R or H3R.

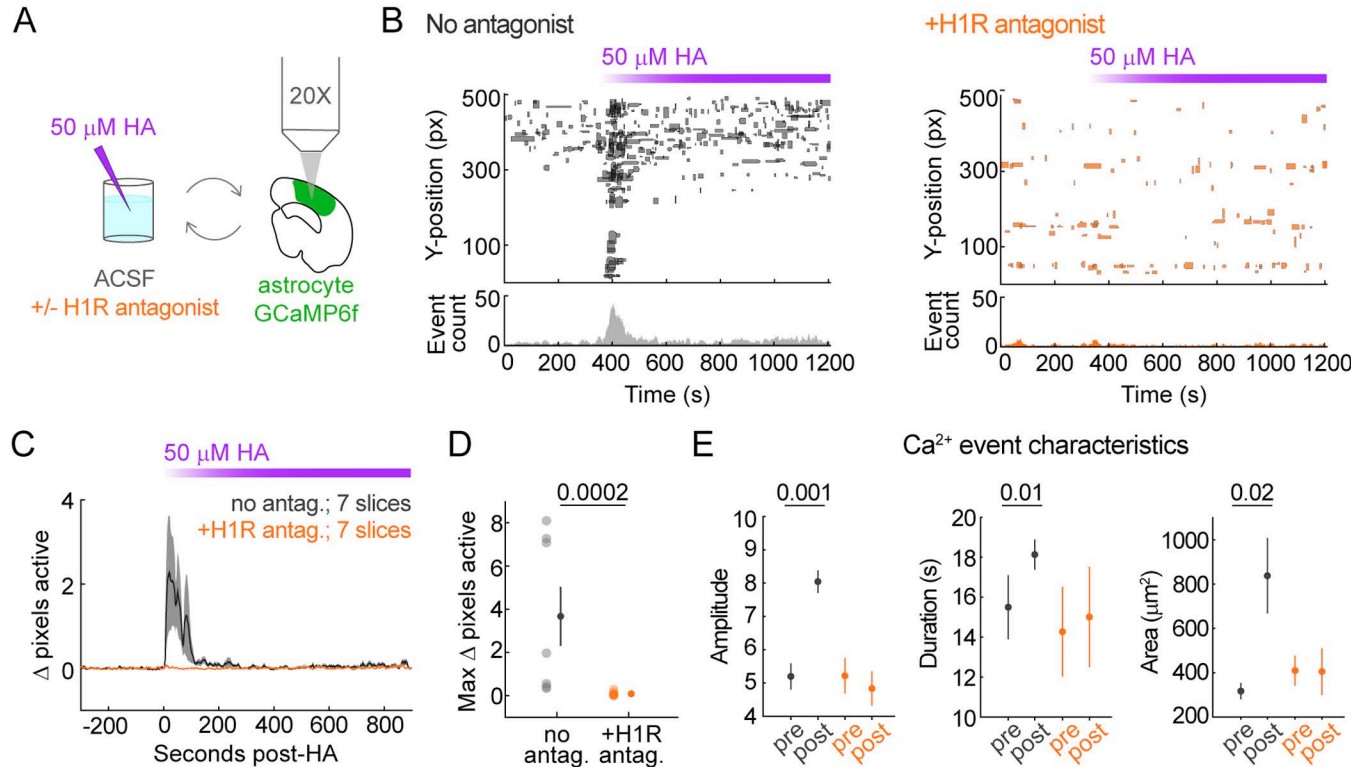

**Fig 2. Histamine (HA)-triggered astrocyte Ca²⁺ is histamine-1-receptor (H1R)-dependent. (A)** Experimental schematic. Acute V1 slices from adult mice (≥P28) expressing cortical astrocytic GCaMP6f are 2P-imaged before and after 50 µM HA addition. **(B)** Y-axis position and duration of Ca²⁺ events before and after 50 µM HA addition without (left, gray) or with 50 µM H1R antagonist chlorpheniramine (right, orange); lower bar plots show number of active events per second. Purple bar = time HA is recirculating. **(C)** Change in % active pixels relative to mean 60 s pre-HA ± H1R antagonist. Traces = mean across slices. Shaded error bars = SEM. **(D)** Summary of data in C. Maximum Δ pixels active during 2-min post-HA for each slice, with mean ± SEM at right (no antagonist: 3.7 ± 1.4, +H1R antagonist: 0.09 ± 0.04). *p*-values via one-sided Wilcoxon rank-sum test. **(E)** Amplitude (left), duration (middle), and area (right) of Ca²⁺ events pre- and post-HA ± H1R antagonist. Mean ± 95% CI estimated via bootstrapping with replacement. *p*-values via permutation test on data before bootstrapping. Underlying data available on Dryad (https://doi.org/10.5061/dryad.2280gb64x); panels B–E: Fig2.zip, Fig2_H1Rpharm_data.mat.

PLOS Biology

## Astrocyte-specific H1R expression is required for HA-triggered astrocyte Ca²⁺ activity

The pharmacological inhibition of H1Rs above was not cell type-specific. To next test whether the HA-triggered Ca²⁺ activity in astrocytes required astrocyte-specific H1R expression, we compared Ca²⁺ responses to HA in WT and H1R cKO astrocytes. Thus, we used neonatal (P0–3) viral injections in H1R$^{fl/fl}$ mice to drive astrocytic Cre and GCaMP6f expression via the GFAP promoter (Fig 3A and 3B; Methods) and measured astrocyte Ca²⁺ activity in acute V1 slices from ≥P28 mice. In this experimental strategy, acute V1 slices contained a mosaic of WT and cKO astrocytes, which facilitated within-slice paired comparisons (S3A and S3B Fig). Astrocyte-specific deletion of H1R in V1 was verified by analyzing colocalization of astrocytic Cre-RFP, the astrocyte marker S100β, and H1R mRNA in fixed tissue via RNAscope (Fig 3C and 3D). Additionally, since neuronal precursors express GFAP [62,63] and mature neurons express H1R [4,6], we ensured that neurons were not transduced with the Cre-RFP virus by quantifying overlap of Cre and NeuN expression (percentage of pixels in RFP⁺ soma that were NeuN⁺, Fig 3E and 3F). On average, 7% of the pixels in individual RFP⁺ soma were NeuN⁺, while 96% of pixels were RFP⁺, with some NeuN and RFP overlap coming from different z-planes (see S2A–S2C Fig). These results demonstrate overall lack of neuronal Cre-RFP expression, which confirms our observation that RFP⁺ cells consistently exhibited astrocytic morphology (Fig 3B, 3C, and 3E).

We next 2P-imaged GCaMP6f dynamics in WT (RFP⁻) and cKO (RFP⁺) astrocytes before and after 50 μM HA addition, and analyzed AQuA-detected Ca²⁺ activity within manually drawn regions of interest (ROIs) around WT and cKO astrocytes (Figs 4A, S3A, and S3B; S3 Video). WT astrocytes exhibited a robust increase in the percent of each ROI active (% ROI active) when stimulated with 50 μM HA. This effect was almost completely abolished in cKO astrocytes (Fig 4B and 4D), which showed a comparatively small increase in % ROI active (~20% in WT versus ~2% in cKO cells; S3C Fig). This modest change in the cKO % ROI active is likely due to an increase in Ca²⁺ event rate in cKO cells (S3E Fig), without a corresponding increase in Ca²⁺ event area (S3D Fig). These results suggest that cKO cells are not directly responsive to HA but maintain some HA-triggered activity via connectivity to the rest of the astrocyte network or in response to activation of neuronal H1R.

To test whether the cKO astrocytes were unresponsive specifically to HA or more generally, we bath-applied 10 μM NE—which drives large astrocyte Ca²⁺ elevations in cortex [21–23,36,37,64]—immediately after the HA application (Fig 4A). The cKO astrocytes showed clear Ca²⁺ increases when NE was bath-applied (Fig 4B and 4D), demonstrating that they retain Ca²⁺ responsiveness and their non-response to HA was specific to this neuromodulator. However, neighboring WT astrocytes exhibited attenuated responses to NE after HA stimulation (Fig 4B and 4D). This could be due to exhaustion of Ca²⁺ stores since we saw no genotype-specific difference in NE-triggered % ROI activated when slices were not previously stimulated with HA (Fig 4C and 4F; S4 Video). However, we think Ca²⁺ store exhaustion is unlikely because NE responses in cerebellar astrocytes recover within 4 min after a single preceding NE stimulation [11], and NE was added to the recirculating ACSF bath at least 10 min after the initial HA addition. Instead, our results could reflect HA-triggered changes in astrocyte activity that outlast acute H1R activation, consistent with the observation that the decrease in NE-triggered WT responses relative to neighboring cKO responses is not correlated with the time between HA and NE addition (Fig 4E).

We tested this idea further by quantifying changes in Δ$F$/$F$ in individual astrocytes stimulated with low [HA] followed by NE (Fig 5A) or with NE only (Fig 5B). In the sequential ligand experiment (Fig 5A), we used 0.5 μM HA, which we have shown drives low levels of astrocyte Ca²⁺ activity (Figs 1 and S1) and therefore should not exhaust Ca²⁺ stores. As expected, we found that WT astrocytes responded to 0.5 μM HA while cKO astrocytes did not (Fig 5C–5E). Both genotypes responded to 10 μM NE, but Ca²⁺ elevations were larger in cKO astrocytes relative to WT (Fig 5C–5E). This does not appear to result from Ca²⁺ store depletion in WT astrocytes, as NE responses were smaller even in WT cells that showed little to no response to 0.5 μM HA (Fig 5D). We found consistent results when we analyzed cell-wide fluorescence changes in slices only stimulated with NE (same dataset as shown Fig 4C and 4F). In this case, cKO astrocytes exhibited larger NE-triggered Ca²⁺ responses as measured by ΔF/F (Fig 5F–5G), confirming that HA-stimulated Ca²⁺ store depletion

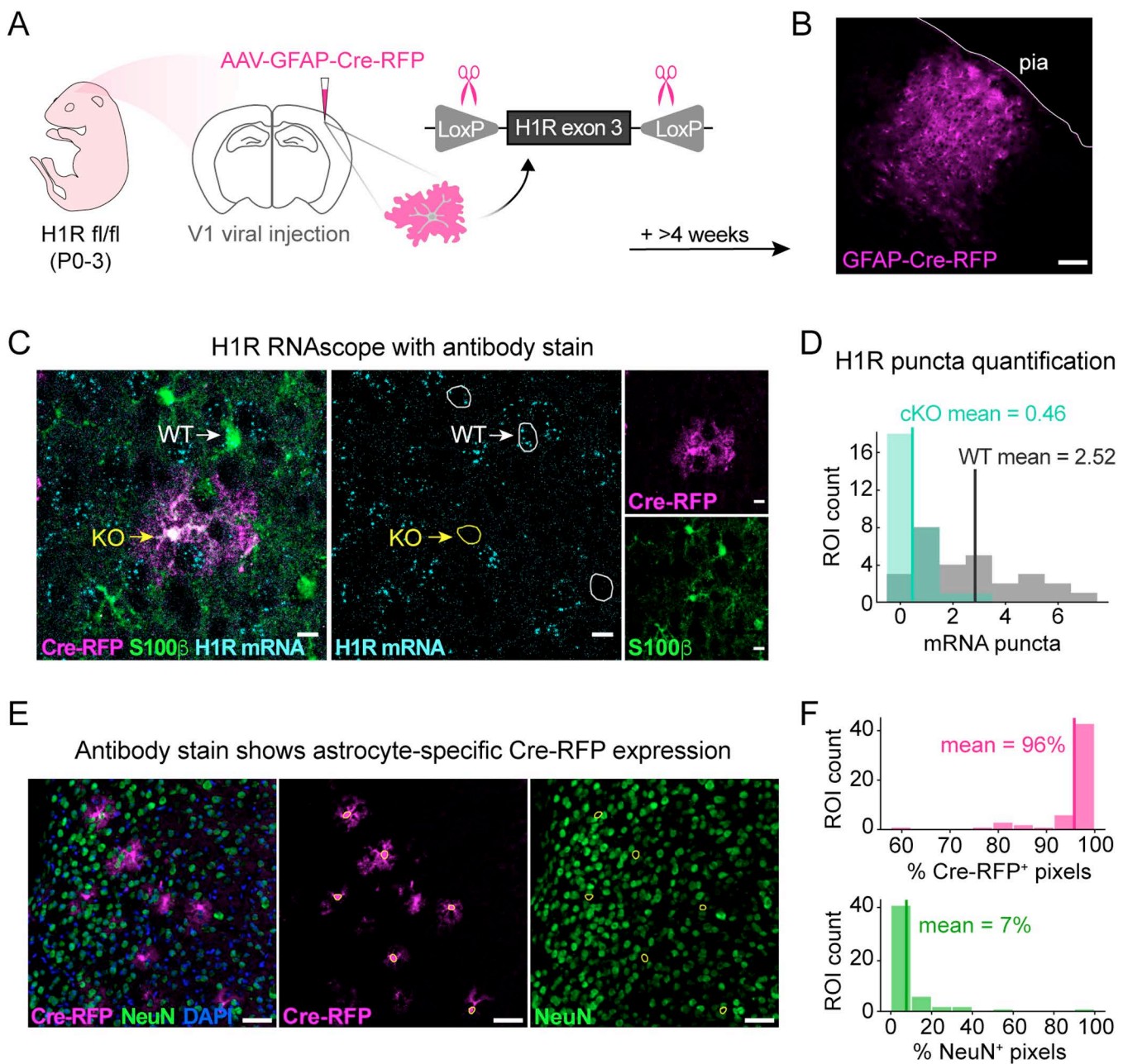

**Fig 3. Astrocyte-specific Cre-virus drives histamine-1-receptor (H1R) deletion specifically in cortical astrocytes. (A)** Schematic illustrates H1R conditional knockout procedure. Astrocytic Cre-RFP expression is driven via neonatal virus injections. Cre-mediated recombination excises exon 3, which encodes H1R. Proceeding experiments are conducted after 4 weeks of viral transduction. **(B)** Example confocal micrograph showing localized Cre-RFP expression in the cortex ~4 weeks after neonatal virus injection. Cre-RFP⁺ cells are morphologically consistent with astrocytes. White line shows pia; scale bar = 100 µm. **(C)** Example confocal micrograph showing astrocyte expression of S100β (green), Cre-RFP (magenta), and H1R mRNA (cyan). Left: summed z-projection of all three confocal channels. Middle: RNAscope detected H1R mRNA with white and yellow arrows indicating example WT (S100β⁺/RFP⁻/H1R⁺) and cKO (RFP⁺/H1R⁻) soma regions of interest (ROIs), respectively. Right: Cre-RFP expression (top, magenta) and S100β expression (bottom, green). Scale bars = 10 µm. **(D)** Quantification of RNAscope detected H1R mRNA puncta in RFP+ and S100β⁺/RFP⁻ astrocytes in H1R^fl/fl cortex. Histogram shows distribution of 28 cKO soma and 28 randomly sampled data points from a total of 48 WT soma. cKO soma (RFP⁺; $n$ = 28 soma from 8 sections; $N$ = 3 mice) had a mean of 0.46 puncta and a 36% probability of >0 H1R puncta. WT soma (S100β⁺/RFP⁻; $n$ = 28 soma from 8 sections; $N$ = 3 mice) had a mean of 2.52 puncta and a 77% probability of >0 H1R puncta. **(E)** Example confocal micrograph showing astrocytic Cre-RFP (magenta), neuronal NeuN (green), and nuclear marker DAPI (blue). Left: summed z-projection of all three confocal channels. Middle: astrocytic

Cre-RFP signal with astrocyte soma ROIs overlaid. Right: NeuN signal with astrocyte soma ROIs overlaid. Scale bars = 50 μm. **(F)** Quantification of Cre-RFP expression in cortical neurons. Histograms show percentage of astrocyte soma pixels that are RFP+ (top, pink) and NeuN+ (bottom, green). Mean percentage of soma pixels that are RFP+ and NeuN+ is 96% and 7%, respectively ($n$ = 57 soma from 6 sections, $N$ = 3 mice). Underlying data available on Dryad (https://doi.org/10.5061/dryad.2280gb64x); panels C–F: Fig3.zip, panels C–D: Fig3_H1R_mRNA_quantification.xlsx, panels E–F: Fig3_CreRFP_NeuN_colocalization.xlsx.

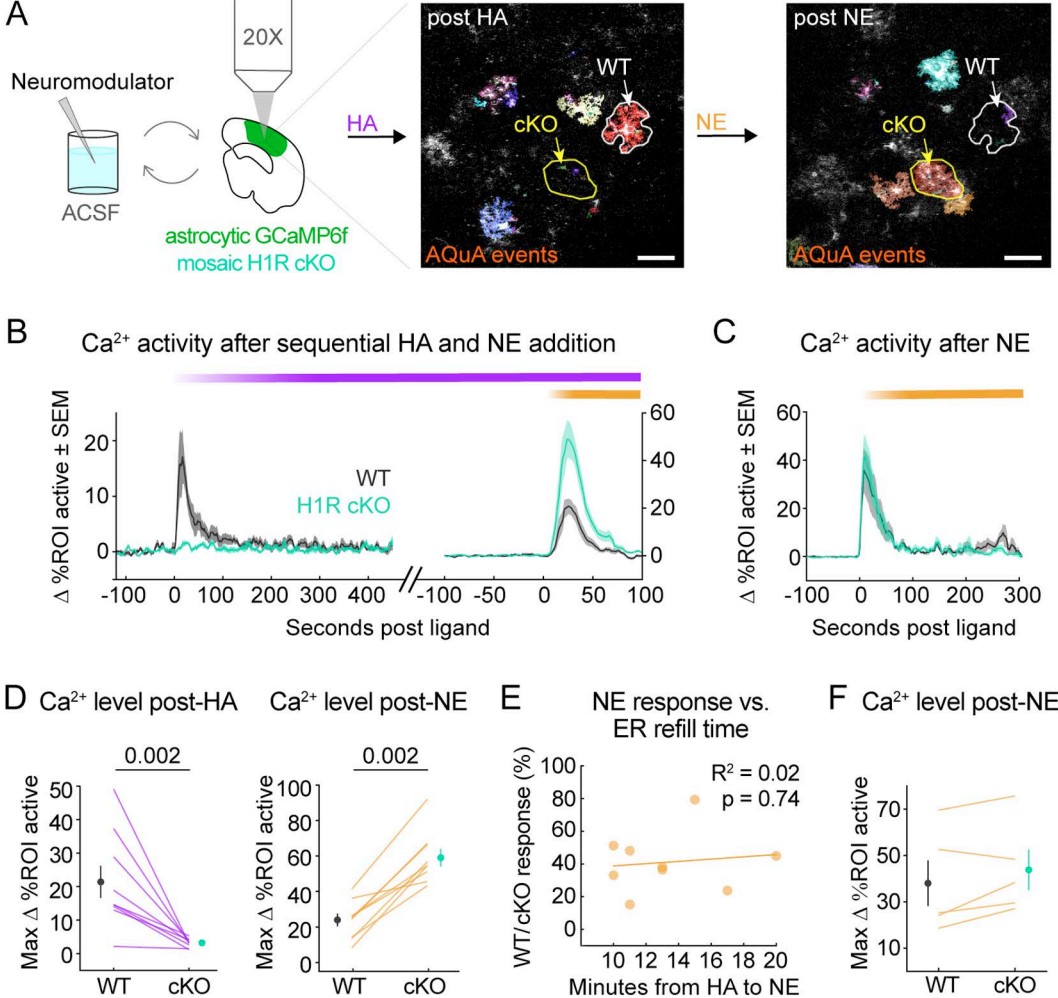

**Fig 4. Histamine (HA) drives astrocyte Ca²⁺ via astrocyte-specific histamine-1-receptor (H1R). (A)** Experimental schematic. Acute V1 slices from H1R^fl/fl mice expressing cortical astrocytic GCaMP6f ± Cre-RFP are 2P-imaged before and after neuromodulator addition. Example images: four summed frames of 2P time-series with overlaid AQuA events after 50 μM HA (left) and 10 μM norepinephrine (NE) (right) addition. Regions of interest (ROIs) and arrows indicate example H1R cKO (yellow arrow) and WT (white arrow) astrocytes (see Methods for identification). Scale bars = 50 μm. **(B)** Mean change in percent of WT or cKO pixels active post-ligand. Purple bar = 50 μM HA addition; yellow bar = 10 μM NE addition. Shaded error bars = SEM. **(C)** Mean change in percent of WT or cKO pixels active after 10 μM NE addition (yellow bar) to untreated slices. Shaded error bars = SEM. **(D)** Summary of data in B. Left: WT vs. cKO mean maximum change in percent-ROI-active post-HA for each slice. Mean across slices ± SEM (WT: 21.4 ± 4.8%; cKO: 3.2 ± 0.4%). Right: post-NE condition (WT: 24.0 ± 3.6%; cKO: 59.1 ± 5.0%). For D and F, $p$-values calculated via one-sided, paired Wilcoxon rank-sum. **(E)** Linear regression analysis showing correlation between WT/cKO NE-response ratio and the amount of time separating HA and NE addition to recirculating ACSF bath. Each data point shows mean WT/cKO response ratio per slice. Displayed $R^2$- and $p$-values indicate no correlation. **(F)** Summary of data in C. Mean maximum change in percent-ROI-active post-NE for each slice. Mean across slices ± SEM (WT: 38.0 ± 9.9; cKO: 43.8 ± 8.8). Data in panels B, D, E collected from 9 slices and 3 mice; data in panels C, F collected from 5 slices and 2 animals. Underlying data available on Dryad (https://doi.org/10.5061/dryad.2280gb64x); panels A–F: Fig 4.zip, panels A–B, D: Fig 4_H1RKO_HA_data.mat, Fig 4_H1RKO_NE_postHA_data.mat, panels C–F: Fig 4_H1RKO_NE_data.mat.

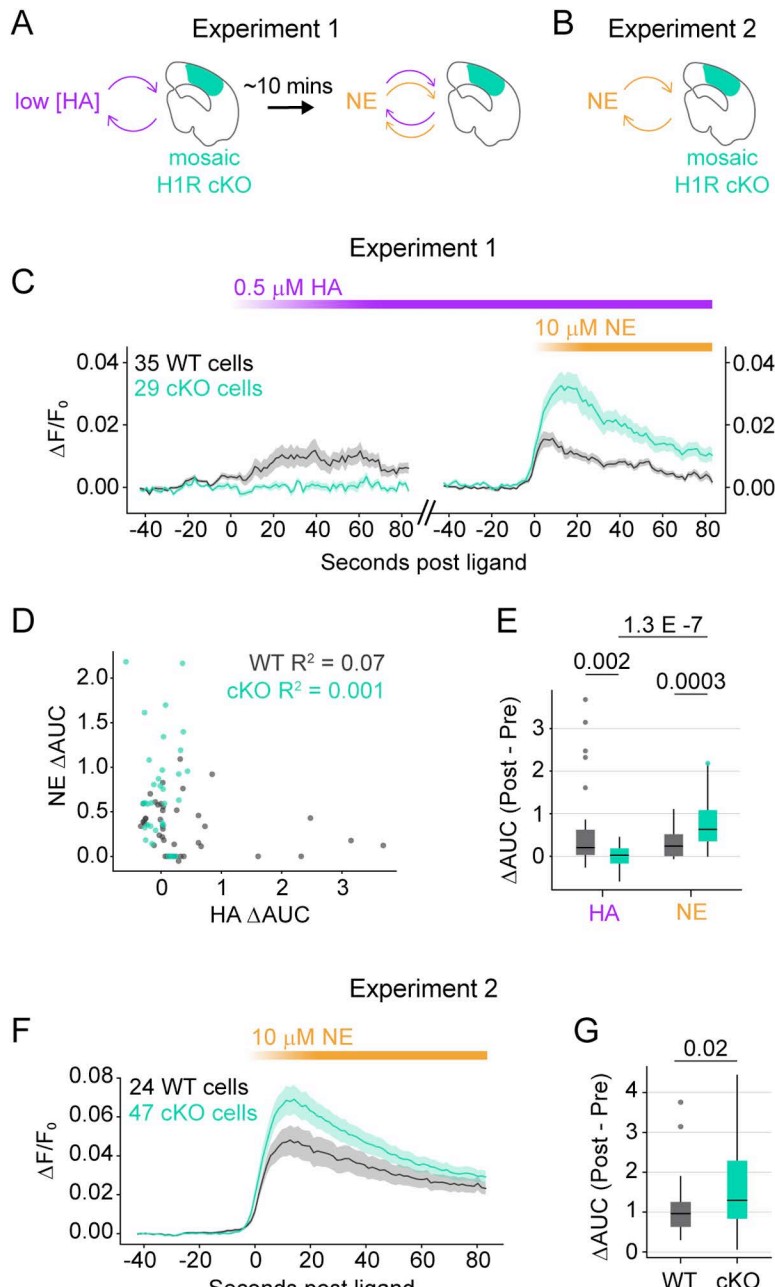

**Fig 5. Histamine-1-receptor (H1R) activity attenuates norepinephrine (NE)-triggered astrocyte Ca²⁺ elevations. (A)** Experiment 1 schematic. Astrocytic Ca²⁺ dynamics were 2P-imaged in a recirculating ACSF bath pre- and post-addition of 0.5 μM HA, followed ~10 min later by 10 μM NE. Results shown in C–E. **(B)** Experiment 2 schematic. Astrocytic Ca²⁺ dynamics were 2P-imaged in a recirculating ACSF bath pre- and post-addition of 10 μM NE. Results shown in F and G. **(C)** ΔF/F for WT and cKO astrocyte regions of interest (ROIs) stimulated with 0.5 μM HA (purple) then 10 μM NE (orange). Traces = mean ΔF/F across ROIs. Shaded error bars = SEM. Genotypic color-code (WT, gray; cKO, cyan) used in panels C–G. **(D)** Linear regression analysis shows relationship between magnitude of HA-triggered and NE-triggered Ca²⁺ responses for cKO (cyan) and WT (gray) ROIs. Ca²⁺ response magnitude is area under the curve (AUC) across post-ligand period (60 s for HA, 30 s for NE) minus AUC across 30 s pre-ligand period. **(E)** Boxplots show normalized AUC (post-pre) for WT and cKO astrocytes stimulated with HA (left) or NE (right). Each box spans interquartile range (25th–75th percentile), with horizontal line indicating median, whiskers extending to most extreme values within 1.5× the IQR, and outliers plotted individually. *p*-values via one-sided Wilcoxon rank-sum test. **(F)** ΔF/F analysis of data shown in Fig 4C and 4F. NE-triggered Ca²⁺ elevations are larger in cKO astrocytes even without previous HA stimulation. Traces = mean ΔF/F across ROIs. Shaded error bars = SEM. **(G)** Boxplots summarizing data in F. AUC during 30 s post-NE normalized to AUC 30 s pre-NE. *p*-value via one-sided Wilcoxon rank-sum test. Data in C–E collected from 8 slices and 3 mice; data in F–G collected from 5 slices and 2 mice. Underlying data available on Dryad (https://doi.org/10.5061/dryad.2280gb64x); panels C–G: Fig 5.zip, panels C–E: Fig 5_H1RKO_NE_postLowHA.mat, panels F–G: Fig 5_H1RKO_NE.mat.

does not drive the genotype-specific differences in NE responses. Together, these results strongly suggest that both stimulated and constitutive H1R activity [65] in astrocytes suppresses their $Ca^{2+}$ responses to NE.

**H1R cKO increases cortical astrocyte $Ca^{2+}$ activity during wakefulness**

We next tested how astrocytic H1R signaling shapes in vivo astrocyte $Ca^{2+}$ dynamics across sleep and wake states. Histaminergic neurons fire tonically during wake and are almost silent during NREM and REM sleep [66], leading to high extracellular HA during wake, low during NREM, and lowest during REM, as shown in prefrontal cortex [67]. We confirmed these state-dependent HA dynamics in V1 via freely moving fiber photometry recordings of extracellularly facing GRAB-histamine (S4 Fig), and thus hypothesized that astrocytic H1R deletion would lead to reduced astrocyte $Ca^{2+}$ activity specifically during the wake state. To test this, we genetically deleted H1R uni-hemispherically in cortical astrocytes by virally co-expressing astrocyte-specific Cre and jRGECO in V1 of adult H1R^fl/fl mice, and implanted an optical fiber directly over the virus injection site (Figs 6A, S5A, and S5B). We also implanted an EEG screw in contralateral V1 to track sleep/wake states (Figs 6A and S5A). Two weeks after the virus injections, we measured astrocyte jRGECO dynamics via fiber photometry across sleep/wake in freely moving mice during their circadian light phase (see Methods). We then used a Hidden Markov model (HMM) to score NREM and wake states and manually classified REM periods, since the automated HMM did not distinguish REM from wake with sufficiently high fidelity (see Methods). Using this scoring method, both genotypes exhibited well-established wake-, NREM-, and REM-specific patterns in EEG dynamics (S6 Fig). Both WT and cKO mice showed elevated delta and reduced theta power during NREM relative to wake and REM, and elevated delta during wake relative to REM (S6B and S6C Fig). Additionally, both WT and cKO mice exhibited increased theta power at NREM-to-REM relative to NREM-to-wake transitions (S6D and S6E Fig).

Since astrocytes respond via $Ca^{2+}$ to multiple wake-specific signals including HA, NE, DA, ACh, and glutamate [68], we predicted that the H1R cKO animals would show an attenuation of wake-specific astrocyte $Ca^{2+}$ activity, rather than a complete blockade. To quantify $Ca^{2+}$ activity, we used a peak-finding function to characterize peak amplitude and frequency during wake, NREM, and REM in WT and cKO mice (Fig 6B). We then plotted $Ca^{2+}$ peak frequency per bout against mean theta/delta ratio, which distinguishes NREM, REM, and wake periods (Fig 6C), to determine how to accurately quantify state-specific peak dynamics. As expected, NREM, wake, and REM exhibited low, medium, and high theta/delta ratios, respectively. Both NREM and wake bouts exhibited $Ca^{2+}$ peaks, although this was true for less than a third of NREM bouts (34/286 WT, 85/344 cKO). $Ca^{2+}$ peaks were rarely detected during REM (fraction of REM bouts with $Ca^{2+}$ peaks: 10/69 WT, 2/114 cKO). Since only a fraction of NREM and REM bouts exhibited any $Ca^{2+}$ peaks, we summarized $Ca^{2+}$ peak dynamics by comparing the per-recording median peak frequency and amplitude (Fig 6D and 6E).

Consistent with published cortical data, both genotypes exhibited increased astrocyte $Ca^{2+}$ peak frequency during wake relative to NREM and REM (Fig 6D). We found that cKO peak frequency during wake was higher than WT (0.44 versus 0.34 peaks/min). In contrast, cKO exhibited lower REM-specific peak frequency than WT (0 versus 0.07 peaks/min), although this effect was driven by just 3/16 WT recordings. Since median amplitude was calculated exclusively from behavioral bouts with a peak frequency >0 and the REM state yielded too few bouts with $Ca^{2+}$ peaks for meaningful statistics—we restricted our amplitude comparison to those occurring during wakefulness and NREM sleep (Fig 6E). $Ca^{2+}$ peak amplitude was higher during wake (3.4 WT; 3.7 cKO) relative to NREM sleep (2.8 WT; 3.1 cKO) with no significant differences between genotypes.

In addition to quantifying astrocyte $Ca^{2+}$ during entire sleep/wake periods, we also measured changes in astrocyte $Ca^{2+}$ across sleep/wake transitions. To do this, we plotted the change in jRGECO fluorescence relative to mean jRGECO fluorescence during the pre-transition period, which was 200 s for wake/NREM transitions and 60 s for REM transitions. Consistent with previous data [18,19,24], both genotypes exhibited a decrease in $Ca^{2+}$ at wake-to-NREM transitions (Fig 6F) and a clear increase at NREM-to-wake transitions (Fig 6G). Surprisingly, since H1R is required for HA-triggered astrocyte $Ca^{2+}$ in slices, we found that the cKO exhibited larger $Ca^{2+}$ elevations at wake-onset, as measured by area under the curve (AUC) during the first

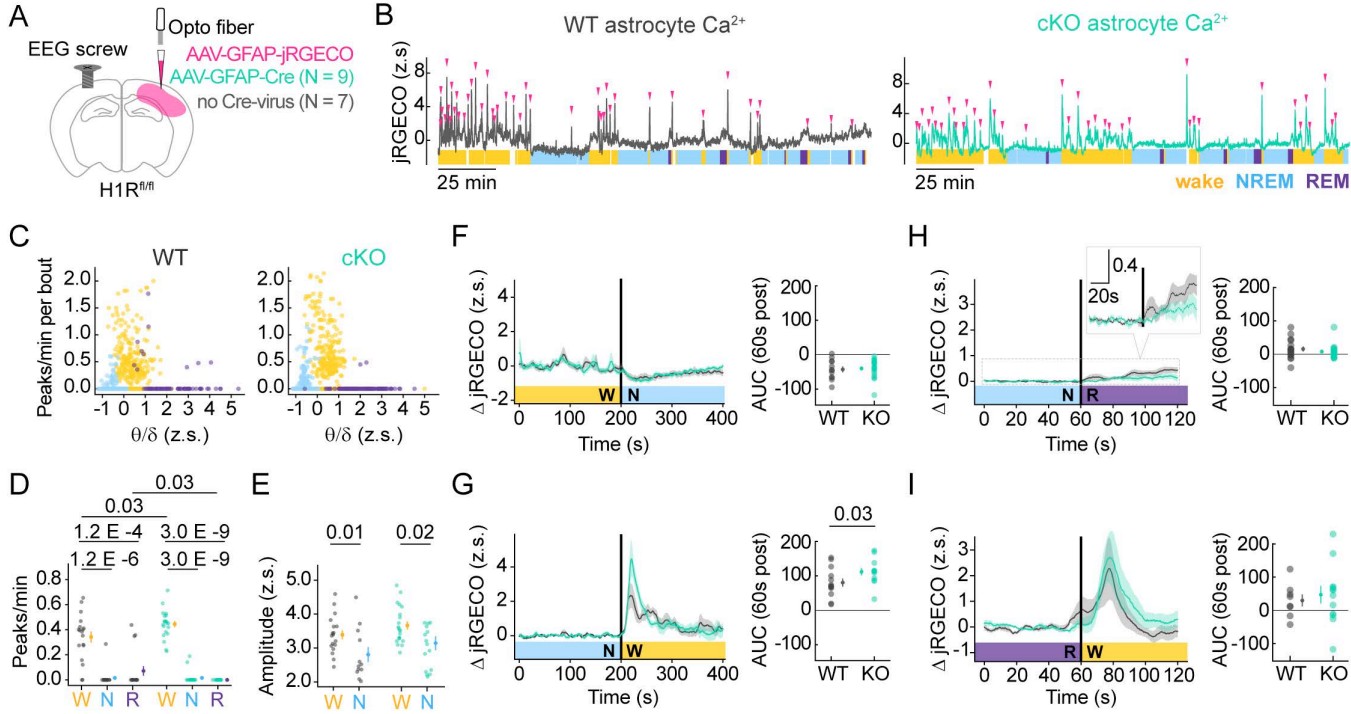

**Fig 6. Astrocytic histamine-1-receptor (H1R) conditional knockout (cKO) specifically disrupts astrocyte Ca²⁺ dynamics during wake.**
**(A)** Conditional H1R knockout mouse surgery schematic shows astrocytic jRGECO and Cre-GFP viral injections into V1, photometry fiber placement, and contralateral EEG screw placement in H1R^{fl/fl} mice. Wild-type surgeries omitted Cre virus. **(B)** Example z-scored jRGECO photometry traces from WT (left, gray) and H1R cKO (right, cyan) mice aligned to color-coded wake (yellow), NREM (blue), and REM (purple) periods with detected jRGECO peaks indicated by magenta arrows. Peaks detected using Python's signal.find_peaks function with prominence level = 2, distance = 5 s, width = 1 s. White blocks in sleep/wake scoring indicate periods that were not successfully scored by HMM. Horizontal scale bar = 25 min. **(C)** Per bout analysis showing jRGECO peak frequency plotted against theta/delta ratio to distinguish NREM (blue), wake (yellow), and REM (purple) bouts in WT (left) and cKO (right). Very few REM bouts exhibit detectable jRGECO peaks. **(D)** Median jRGECO peaks per minute during wake, NREM, and REM in WT and H1R cKO recordings. Average median ± SEM at right. p-values calculated via one-sided Wilcoxon rank-sum test. **(E)** Median jRGECO peak amplitudes during wake and NREM in WT and H1R cKO recordings. Average median ± SEM at right. p-values calculated via one-sided Wilcoxon rank-sum test. **(F–I)** Event-triggered averages of change in jRGECO relative to mean of pre-transition period. Traces = mean jRGECO activity. Shaded error bar = SEM. Righthand plots show mean area under the curve (AUC) following state transition for WT and cKO. AUC is integration of signal during 60 s post-transition. Data points show mean per recording with overall mean ± SEM. p-values calculated via one-sided Wilcoxon rank-sum test. **(F)** Wake-NREM transitions spanning 400 s. **(G)** NREM-wake transitions spanning 400 s. **(H)** NREM-REM transitions spanning 120 s. Inset shows zoom-in around y-axis. **(I)** REM-wake transitions spanning 120 s. Underlying data available on Dryad (https://doi.org/10.5061/dryad.2280gb64x); panels B–I: Fig 6_WT_grabHA. pkl, Fig 6_KO_jRGECO.pkl, Fig 6_Fig 7_WT_grabAd.pkl, Fig 6_Fig 7_KO_grabAd.pkl.

60 s of wakefulness (Fig 6G). We did not observe any genotype-specific differences in Ca²⁺ activity around NREM-to-REM (Fig 6H) or REM-to-wake transitions (Fig 6I). Finally, in contrast to that observed in somatosensory and prefrontal cortices [18,19], we found that Ca²⁺ levels increased slightly at NREM-to-REM transitions (inset, Fig 6H). This increase was modest when compared to Ca²⁺ dynamics at NREM-to-wake and REM-to-wake transitions, suggesting that V1 astrocytes exhibit some Ca²⁺ activity during REM, or at least at REM-onset, which might be better observed via imaging methods with greater spatial resolution.

Overall, we found that H1R cKO animals exhibited more Ca²⁺ activity during wake (Fig 6D and 6G) when extracellular HA levels are high [53,58,67,69], a counterintuitive result, considering HA-triggered Ca²⁺ requires H1R expression (Fig 4B and 4D). However, this result is consistent with our ex vivo results demonstrating larger NE-triggered Ca²⁺ responses in cKO astrocytes relative to WT (Fig 5), suggesting that H1R activity in astrocytes modulates how these cells respond to other wake-specific signals.

## Astrocyte-specific H1R activity modulates REM-specific extracellular adenosine dynamics in cortex

Astrocytes can increase cortical synchrony, a marker of reduced arousal, by promoting adenosine-mediated neuronal inhibition [34,38,70,71]. Further, neuromodulatory input may play key roles in shaping astrocyte-adenosine signaling [37–39]. Thus, we compared extracellular adenosine levels in V1 of WT and H1R cKO mice using fiber photometry recordings of the extracellularly facing fluorescent adenosine sensor, GRAB-Adenosine (GRAB-Ado) [55], while simultaneously tracking sleep/wake states via an EEG screw in contralateral V1 (Fig 7A and 7B).

For both WT and cKO animals, GRAB-Ado levels in cortex were high during wake and low during NREM and REM sleep according to the distribution of z-scored GRAB-Ado fluorescence (Fig 7B and 7C), consistent with measurements in basal forebrain using GRAB-Ado [55,72]. Differences between the two genotypes were observed during REM sleep, with cKO mice exhibiting lower overall GRAB-Ado levels relative to WT (Fig 7C). In cKO mice only, GRAB-Ado during REM was also lower than NREM GRAB-Ado (Fig 7C).

We next quantified GRAB-Ado dynamics around NREM/wake transitions that spanned 400 s and REM transitions that spanned 60–120 s. As expected, GRAB-Ado increased at NREM-to-wake (Fig 7D) and decreased at wake-to-NREM (Fig 7E) with no differences between genotypes, as shown by comparison of AUC during the 200 s post-transition period (Fig 7D and 7E, right). We did observe distinct WT and cKO GRAB-Ado dynamics at REM transitions. While both genotypes

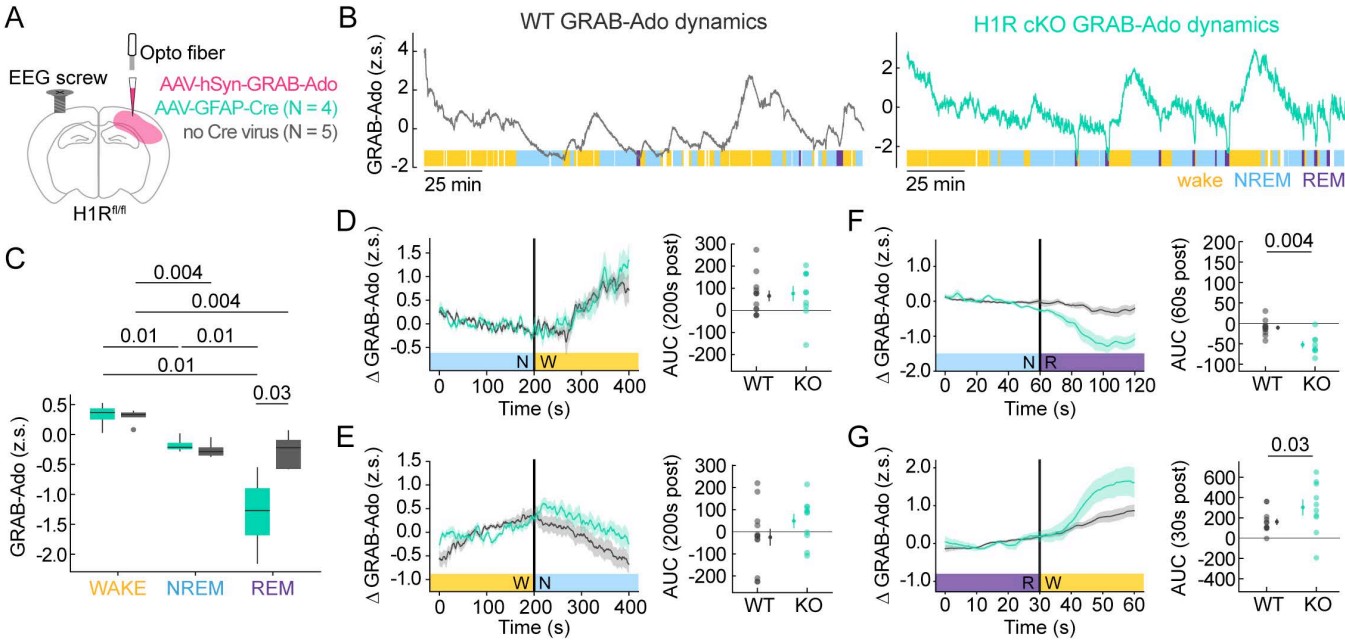

**Fig 7. Astrocytic histamine-1-receptor (H1R) activity shapes REM-specific extracellular adenosine dynamics. (A)** Conditional H1R knockout mouse surgery schematic shows extracellular GRAB-Ado and astrocytic Cre viral injections into V1, photometry fiber placement, and contralateral EEG screw placement in H1R^fl/fl mice. WT surgeries omitted Cre virus. H1R cKO N=4; WT N=5. **(B)** Example z-scored GRAB-Ado photometry traces from WT (gray) and H1R cKO (cyan) recordings aligned to color-coded wake (yellow), NREM (blue), and REM (purple) periods. White blocks in sleep/wake scoring indicate periods that were not successfully scored by HMM. Horizontal scale bar = 25 min. **(C)** Distribution of GRAB-Ado z-score values from wake, NREM, or REM in WT and cKO recordings. Each box spans interquartile range (25th–75th percentile), with horizontal line indicating median, whiskers extending to most extreme values within 1.5× the IQR, and outliers plotted individually. p-values via one-sided Wilcoxon rank-sum test. **(D–G)** Event-triggered averages of change in GRAB-Ado relative to mean of pre-transition period. Trace = mean across recordings. Shaded error bars = SEM. Right: mean area under the curve (AUC) following state transition for WT and cKO. AUC is integration of GRAB-Ado signal during post-transition period. Data points show mean per recording with overall mean ± SEM. p-values calculated via one-sided Wilcoxon rank-sum test. **(D)** NREM-wake transitions spanning 400 s. **(E)** Wake-NREM transitions spanning 400 s. **(F)** NREM-REM transitions spanning 120 s. **(G)** REM-wake transitions spanning 60 s. Underlying data available on Dryad (https://doi.org/10.5061/dryad.2280gb64x); panels B–G: Fig 6_Fig 7_WT_grabAd.pkl, Fig 6_Fig 7_KO_grabAd.pkl.

exhibited GRAB-Ado decreases at NREM-to-REM transitions, the cKO mice exhibited larger decreases in GRAB-Ado levels (Fig 7F) as shown by comparison of AUC during the 60 s post-transition period (Fig 7F, right). For REM-to-wake, we analyzed transitions spanning 60 s (Fig 7G) because many wake bouts that followed REM were less than 60 s. We found that cKO mice exhibited a larger increase in GRAB-Ado at wake-onset following REM relative to WT as measured by comparison of AUC during 30 s post-transition period (Fig 7G, right).

Overall, we found that extracellular adenosine levels in cortex increased throughout wake and decreased during NREM, consistent with previous data [30]. We also found that extracellular adenosine transiently decreased further with REM-onset. Our results show that astrocytic H1R activity narrows the dynamic range of extracellular adenosine fluctuations specifically across REM sleep transitions, suggesting that activation of astrocytic H1R during wake may have effects on astrocyte physiology beyond the immediate timeframe of HA release and into subsequent sleep periods.

## Astrocytic H1R deletion in cortex shapes wake and REM sleep dynamics

HA promotes arousal via H1R activation, and astrocyte-specific H1R contributes to the wake-promoting effects of this receptor when knocked out brain-wide [54]. However, whether cortical astrocytic H1R deletion is sufficient to shape cortical neuronal dynamics underlying sleep/wake has so far remained untested. In a separate experiment from the photometry experiments above, we targeted genetic deletion of astrocytic H1R to V1 and examined cortical arousal and sleep/wake behavior. To do this, we injected either an astrocyte-specific, GFP-tagged Cre or a sham virus into V1 of H1R^fl/fl mice to generate H1R cKO and negative control mice (WT mice), respectively. An EEG screw was implanted directly over the virus injection site to track cortical dynamics at the site of astrocytic H1R deletion (Figs 8A and S5C and S5D) across sleep/wake recorded during the animal's circadian light phase (see Methods). We found viral transduction localized below the EEG screw, with Cre or sham virus transduction evident across cortex and in hippocampus, but not in deeper subcortical regions that regulate sleep/wake dynamics (S5C–S5H Fig).

Using a HMM with manual REM scoring as in photometry experiments (see Methods), almost the entirety of each recording was classified as wake, NREM, or REM (average of 96.0% [WT] and 96.2% [cKO] of each recording scored). Considering the wake-promoting effects of H1R, we were surprised to find that cKO mice spent more time awake compared to WT, 67.3% and 59.3% time awake, respectively (Fig 8B). We also found that cKO spent less time in REM sleep: 1.71% versus 3.56% in WT (Fig 8B). This genotype-specific difference in REM sleep time coincided with decreased REM bout frequency; cKO mice exhibited 0.90 bouts/hour and WT exhibited 1.42 bouts/hour (Fig 8C), but no significant difference in REM bout durations (Fig 8D, WT = 1.58 min, cKO = 1.15 min). The two genotypes also showed similar individual bout durations for wake and NREM states, with means of 9.10 and 6.34 min, respectively (Fig 8D).

We next assessed cortical frequency band dynamics across sleep/wake. When comparing power spectral densities (PSDs), frequencies spanning 1–20 Hz were similarly represented in WT and cKO mice during wake, NREM, and REM periods (Fig 8E). We further quantified cortical dynamics by comparing percent power in the delta (1–4 Hz), theta (6–10 Hz), and sigma (10–15 Hz) frequency bands relative to total power in wake, NREM, or REM. We saw no genotype-specific differences in band power during wake, NREM, or REM (Fig 8F). Together, these data demonstrate that astrocyte-specific H1R deletion localized near an EEG electrode in V1 increased wake time and decreased REM sleep time, with minimal additional effects on sleep/wake dynamics.

We found a slightly different sleep/wake phenotype in cKO photometry mice that had H1R deleted in the contralateral hemisphere from the EEG screw (S5A, S5B, and S6A Figs). First, these contralateral cKO mice did not exhibit changes in percent time in wake, NREM, or REM states relative to WT (S7B Fig) and they did not exhibit the decrease in REM bout frequency that we saw in ipsilateral cKO mice (Fig 8C). Instead, contralateral cKO mice exhibited fewer wake bouts per hour relative to their WT counterparts (S7C Fig). They also exhibited shorter REM and longer wake bout durations relative to WT (S7D Fig). While these results are distinct from ipsilateral results, ipsilateral cKO data did exhibit these trends (less frequent wake bouts, shorter REM bouts, and longer wake bouts) but the comparisons did not reach statistical

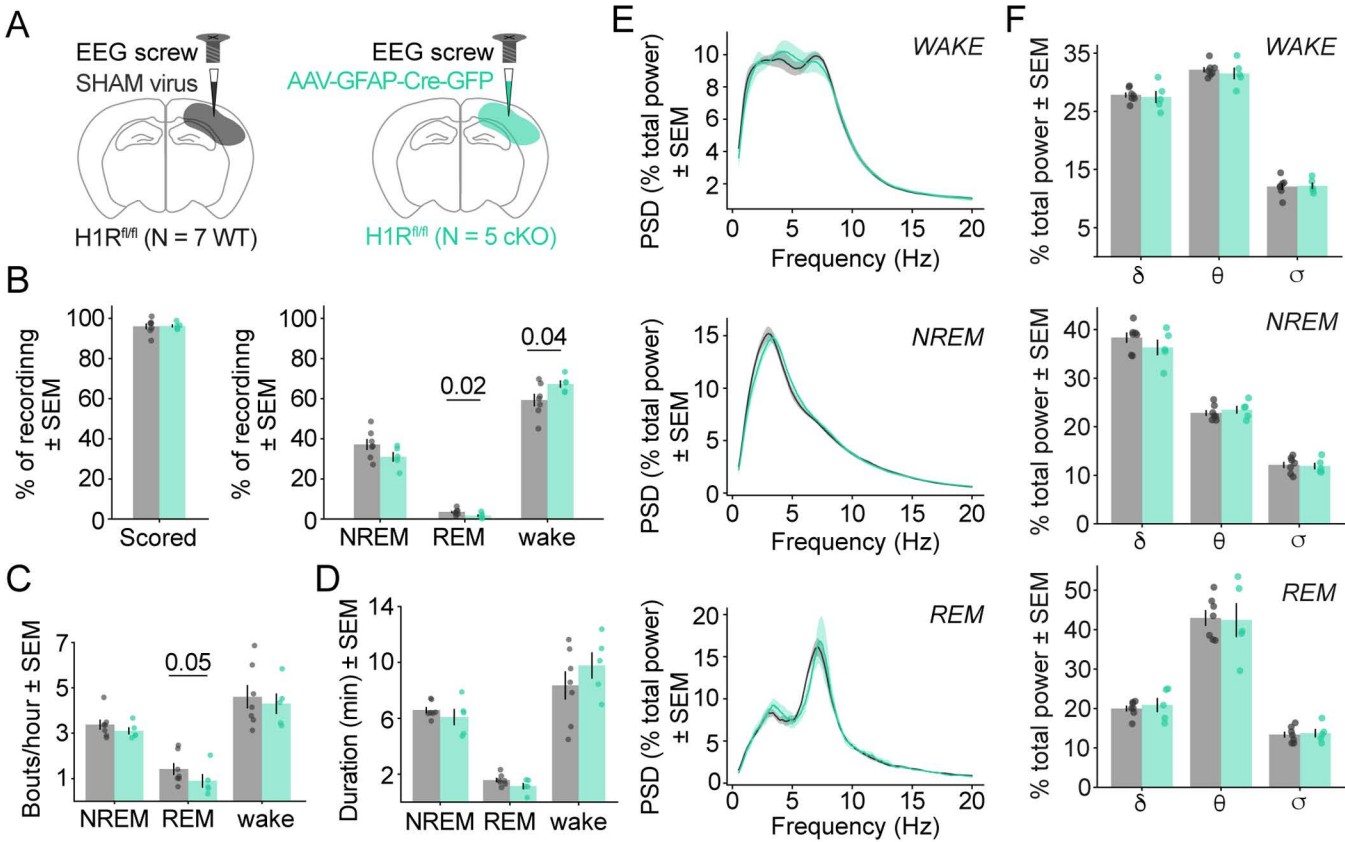

**Fig 8. Uni-hemispheric cortical astrocyte-specific histamine-1-receptor (H1R) conditional knockout (cKO) changes wake and REM dynamics with minimal impact to other sleep/wake features. (A)** Surgery schematics show V1 astrocytic Cre or sham virus injection and EEG screw location in transgenic H1R$^{fl/fl}$ mice. Mouse number in each cohort listed below schematics. Color code for experimental cohorts used in B–F. **(B–D)** WT and cKO sleep/wake architecture. For each plot, overlaid data points show mean per mouse. Error bars = SEM. *p*-values via one-sided Wilcoxon rank-sum test. **(B)** Left: % individual recordings scored as wake, NREM, or REM (WT: 96.0 ± 1.5%, cKO: 96.2 ± 0.7%). Right: % time in wake, NREM, and REM (mean for WT: 59.3 ± 3.2 wake, 37.2 ± 2.7 NREM, 3.6 ± 0.6 REM; cKO: 67.3 ± 1.9 wake, 31.0 ± 2.4 NREM, 1.7 ± 0.6 REM). cKO mice spend significantly more time awake and less time in REM sleep than WT. **(C)** Bouts per hour for each behavioral state (mean for WT: 4.6 ± 0.5 wake, 3.4 ± 0.2 NREM, 1.4 ± 0.2 REM; cKO: 4.3 ± 0.5 wake, 3.1 ± 0.2 NREM, 0.9 ± 0.3 REM). REM bout frequency significantly lower in cKO relative to WT. **(D)** Bout duration in minutes for each behavioral state (mean for WT: 8.4 ± 1.0 wake, 6.6 ± 0.2 NREM, 1.6 ± 0.2 REM; cKO: 9.8 ± 1.0 wake, 6.1 ± 0.6 NREM, 1.2 ± 0.2 REM). **(E)** Power spectral density (PSD) plots show WT and cKO % power relative to total power across 1–20 Hz during wake, NREM, and REM. PSDs calculated via multitaper spectrogram. Traces = mean across mice; shaded error bar = SEM. **(F)** Bar plots show WT and cKO % band power (relative to total power in each state) for δ (1–4 Hz), θ (6–10 Hz), and σ (10–15 Hz) during wake (top), NREM (middle), and REM (bottom). Bars show overall mean and overlaid data points show mean per mouse. Error bars = SEM. One-sided Wilcoxon rank-sum test with Benjamini–Hochberg multiple comparisons correction does not detect any significant differences between WT and cKO. Underlying data available on Dryad (https://doi.org/10.5061/dryad.2280gb64x); panels B–F: Fig 8_ipsi_SHAM.pkl, Fig 8_ipsi_KO.pkl.

significance as measured by a one-sided Wilcoxon rank-sum test (compare Figs 8C and S7C; Figs 8D and S7D). This could be due to smaller sample sizes in the ipsilateral cohort (*N* = 7 WT, 5 cKO) compared with the contralateral cohort (*N* = 7 WT, 9 cKO). Alternatively, the sleep/wake differences between the ipsi- and contralateral cohorts, including the relative sleepiness of the contralateral cohort (compare % time in NREM in Figs 8B and S7B), could be explained by the additional photometry fiber cable on the head mount of mice in the contralateral cohort, which could add physical strain and lead to less activity and increased likelihood of sleep. Recording time of day does not explain differences since recording start times are similarly represented across WT and cKO groups in both the ipsilateral and contralateral cohorts, with 60% of all WT recordings and 71% of all cKO recordings starting before 9.0 ZT and the remaining recordings starting after 9.0 ZT.

In summary, the contra- and ipsilateral cKO mice exhibit a similar overall phenotype (more wakefulness, less REM sleep), although there are differences in the underlying sleep/wake architecture recorded in these distinct experimental cohorts.

## Discussion

Using 2P imaging, pharmacology, astrocyte-specific genetic manipulations, in vivo electrophysiology, and fiber photometry, we have demonstrated that cortical astrocytes integrate histaminergic input via H1R, which leads to cellular- and network-level changes that persist beyond the wake state when HA is released. We have shown that H1R is not only required for cell-autonomous, HA-induced $Ca^{2+}$, but also modulates NE-induced $Ca^{2+}$, broader wake-induced $Ca^{2+}$, and REM-specific extracellular adenosine dynamics in cortex. Astrocytic H1R in cortex also reduces wakefulness and promotes REM sleep, but minimally affects additional sleep/wake dynamics. We propose that histaminergic tone during wake drives H1R signaling in astrocytes, which recalibrates how these cells respond to later neuromodulatory cues. Such minute-to-hour integration may represent a general mechanism by which astrocytes bridge fast amine release with the slower homeostatic processes that shape arousal.

### HA directly activates cortical astrocytes via H1R

Astrocytes are poised to propagate neuromodulatory signals across cortex, as they respond to these signals with $Ca^{2+}$ elevations and form continuous, gap junctionally coupled networks that facilitate widespread regulation of neurons [68,73,74]. Yet, whether neuromodulators directly activate astrocytes, rather than indirectly via neurons, remains somewhat underexplored. We show here that HA drives dose-dependent $Ca^{2+}$ activity in cortical astrocytes via astrocyte-specific H1R, adding HA—alongside NE—to the list of neuromodulators that directly activate astrocytes to sense and regulate arousal [17,73,75]. Further astrocyte-specific KO studies will help round out the still-incomplete picture of which neuromodulatory inputs astrocytes are specifically tuned to, and how these inputs distinctly regulate astrocyte function.

One clue to how neuromodulatory inputs may differentially affect astrocyte activity are the specific spatiotemporal patterns of $Ca^{2+}$ triggered by different neuromodulators. Here, H1R-dependent $Ca^{2+}$ activity was largely confined to individual astrocytes (S3 Video), with limited intercellular propagation, indicating a spatially localized response. However, we did observe increased $Ca^{2+}$ event rate in both WT and H1R cKO astrocytes after HA application, suggesting that gap junctions, or autocrine signals like ATP release, may communicate H1R activity across the astrocyte syncytium. Through this kind of mechanism, focal HA release from histaminergic axons in vivo could cause spatially confined $Ca^{2+}$ elevations in H1R-expressing astrocytes, while also driving a network-wide increase in astrocyte $Ca^{2+}$ event rate. These spatially distinct modes of astrocyte $Ca^{2+}$ could facilitate both local astrocytic regulation of synaptic activity near HA release sites and broader regulation of circuit activity.

Our results also indicate that H1R, which exhibits constitutive activity [65], can shape NE-triggered astrocyte $Ca^{2+}$ responses, even many minutes after initial HA stimulation and in the absence of HA stimulation (Figs 4B, 4D, and 5). Alternatively, repeated stimulation of $Ca^{2+}$ activity, independent of H1R-specific signaling, could explain the genotype-specific differences in NE responses (Fig 4). This idea could be mechanistically tested by comparing WT and cKO responses to consecutive stimulation of non-histaminergic Gq-coupled GPCRs. That said, our interpretation that H1R signaling modulates NE-triggered astrocyte $Ca^{2+}$ is consistent with recent work showing that a single neuromodulator gates responses to other neuromodulators [42], underscoring the synergistic effects of distinct neuromodulators on astrocyte physiology.

### H1R signaling modulates astrocyte activity in vivo

Astrocytes exhibit highly heterogeneous $Ca^{2+}$ activity across arousal states, with increased activity during wake compared to sleep that is driven, at least in part, by noradrenergic input [21–23]. Despite the expression of many types of neuromodulatory GPCRs in astrocytes [4,5], how non-adrenergic neuromodulators regulate astrocytic $Ca^{2+}$ dynamics specific to wake, NREM, and REM remains largely unknown. Our understanding of REM-specific astrocyte $Ca^{2+}$ dynamics is

particularly limited, with $Ca^{2+}$ increasing at NREM-to-REM in basal forebrain [72], but decreasing in somatosensory [18] and frontal cortices [19].

Here, astrocyte-specific H1R deletion altered in vivo $Ca^{2+}$ dynamics, establishing that HA directly activates astrocytes in behaving animals. One caveat to these results is that the viral Cre/lox strategy used to conditionally delete H1R—which allowed specific targeting of cortical astrocytes and exclusion of neuronal Cre expression [34,76–79]—resulted in jRGECO photometry signals from a mixture of WT and cKO astrocytes. Therefore, we suggest that our results are an underestimate of the full extent to which H1R modulates astrocyte $Ca^{2+}$ in vivo. Despite this mosaic H1R deletion, we still observed significant changes in astrocyte $Ca^{2+}$ activity during specific sleep/wake stages. Given our ex vivo data showing that H1R deletion reduces astrocyte $Ca^{2+}$ responsiveness to HA, it was notable to find that H1R cKO mice displayed *increased* $Ca^{2+}$ peak frequency during wake and larger $Ca^{2+}$ elevations at NREM-to-wake transitions, when histaminergic neurons are active and HA is released [52,66,67]. These changes suggest that H1R activity shapes astrocytic responses to non-histaminergic inputs, such as NE or ACh, consistent with our ex vivo data (Fig 5) and with evidence that prior astrocyte $Ca^{2+}$ activity or GPCR stimulation influences ongoing $Ca^{2+}$ dynamics in astrocytes [42,57,80]. Constitutive H1R activity likely plays a significant role in modulating GPCR signaling in astrocytes, since our ex vivo data demonstrate that H1R expression attenuates NE-triggered astrocyte $Ca^{2+}$ even before stimulating cortical slices with HA.

Contrary to data from other cortical areas [18,19], both the WT and cKO animals exhibited a small increase in astrocyte $Ca^{2+}$ at NREM-to-REM transitions when measured during the circadian light phase (Fig 6H). This discrepancy aligns with evidence that astrocyte $Ca^{2+}$ dynamics can differ among cortical areas, for example when examining locomotion-induced $Ca^{2+}$ in V1 versus prefrontal cortex [81]. Our results also align with a report of higher neuronal activity in V1 during REM sleep relative to somatosensory and motor cortices [82]. In this context, our results suggest that H1R signaling might shape REM-specific astrocyte $Ca^{2+}$ activity driven by a combination of ACh and local glutamate released by REM-active neurons in V1, and supports an emerging appreciation for astrocyte heterogeneity across the brain. However, it is possible that brain hemodynamics contribute to the increase in astrocytic jRGECO fluorescence that we observed at REM-onset. Oxygenated hemoglobin absorbs less 560 nm light than 405 nm, so it is formally possible that increased oxygenation at REM-onset leads to over-correction of the 560 nm signal, since negative deflections occurring at REM-onset in the 405 nm signal are subtracted from both 560 and 465 channels (see Methods).

Nevertheless, our results call into question the assumption that sleep- and wake-specific astrocyte $Ca^{2+}$ dynamics are homogeneous across the cortex and indicate that H1R activity modulates astrocyte $Ca^{2+}$ responses to other wake-specific inputs, like NE, ACh, or glutamate.

## Astrocytic H1R signaling modulates extracellular adenosine dynamics across sleep/wake

Astrocytes can regulate neocortical neuronal dynamics via adenosine signaling [83], and can increase extracellular adenosine in response to NE [38–41], suggesting that neuromodulators may stimulate changes in astrocyte-adenosine signaling to modulate arousal. We therefore measured extracellular adenosine dynamics across sleep/wake in WT and H1R cKO cortex using GRAB-Ado. Our results confirm what has been observed in microdialysis studies, where extracellular adenosine in cortex increases across wake periods and falls during NREM sleep [30]. Since microdialysis is limited in temporal resolution, it was previously unclear how adenosine levels changed during relatively short REM periods [30]. We show here that extracellular adenosine markedly drops at NREM-to-REM transitions and begins to increase just prior to wake onset following REM (Fig 7F and 7G), thereby revealing a more complete picture of extracellular adenosine fluctuations in cortex across sleep/wake.

Astrocytic H1R deletion altered REM-specific adenosine dynamics. Astrocytic H1R deletion increased the relative drop and rise in extracellular adenosine at the NREM-to-REM and REM-to-wake transitions, respectively. Thus, H1R cKO mice exhibited a larger dynamic range for adenosine fluctuations around REM transitions. However, because REM transitions are accompanied by large shifts in blood flow [84,85]—and both H1R [86] and astrocytes [87] influence cerebral

hemodynamics—genotypic differences in vascular dynamics may partially contribute to the REM-specific fluctuations in GRAB-Ado fluorescence that we observed. For example, astrocytic H1R deletion could reduce blood flow around REM transitions, which could explain the larger changes in cKO GRAB-Ado fluorescence. This effect could arise from signal normalization via subtraction of the 405 nm signal, which is sensitive to hemodynamic changes due to strong hemoglobin absorption at this wavelength.

Other possible explanations for the REM-specific GRAB-Ado findings here: either wake-driven H1R activation induces lasting astrocytic signaling changes that regulate extracellular adenosine levels, or constitutive H1R activity during REM directly modulates astrocytic regulation of adenosine levels. Although long-term or constitutive GPCR signaling is rarely explored in the astrocyte field, our findings indicate these explanations warrant further investigation. Moreover, as both HA and NE [38–41] have now been linked to astrocytic adenosine signaling, they provide a natural framework for dissecting the distinct and overlapping molecular mechanisms by which GPCR signaling pathways regulate extracellular adenosine. Possible astrocytic mechanisms for regulating extracellular adenosine include $Ca^{2+}$-dependent exocytosis of ATP [88], expression of adenosine kinase [34] and ecto-nucleases, and transport of ATP/adenosine via transmembrane proteins [29,88]. Since we observed minutes-long H1R-dependent effects on astrocytic adenosine regulation, follow-up investigations might focus on signaling pathways that operate on a similar timescale such as $Ca^{2+}$-mediated changes in gene expression or phosphorylation of transporters and enzymes known to regulate ATP/adenosine concentrations. Alternatively, investigating how constitutive or stimulated H1R activity regulates other sleep/wake-sensitive GPCR signaling pathways could lead to insights into how astrocytes affect state-dependent changes in adenosine levels.

### Astrocytic H1R signaling in V1 decreases wakefulness and promotes REM sleep

In contrast to known wake-promoting effects of H1R, we found that astrocyte-specific H1R deletion leads to more time awake and less time in REM sleep. Attenuation of astrocytic H1R signaling throughout the brain previously reduced arousal during wake and did not change time awake or REM sleep dynamics [54]. Our differing results may be explained by the fact that we targeted astrocytic H1R deletion specifically to the cortex, while in previous studies H1R was deleted brain-wide, including subcortically. Thus, our finding suggests that astrocytic H1R signaling specifically in the cortex distinctly regulates sleep/wake, ultimately decreasing wakefulness and promoting REM sleep. These findings are consistent with recent work showing that cortex can regulate sleep/wake behavior [89] and with the sleep-promoting effects of GPCR signaling in cortical astrocytes [24], and suggest that H1R in cortical astrocytes and neurons differentially shapes sleep/wake. Alternatively, as H1R is linked to anxiety regulation [54,90,91], increased wakefulness and reduced REM sleep could reflect increased anxiety in cKO mice rather than changes to sleep/wake architecture regulation. This possibility could also explain sleep/wake differences observed between ipsi- and contralateral cohorts. Elevated anxiety in H1R cKO mice could reduce stress tolerance, disproportionately affecting sleep/wake behavior in photometry animals that experience greater physical strain from an added head mount cable. Future studies will be needed to disentangle how astrocyte-neuromodulatory signaling shapes the interplay between anxiety and sleep behavior.

Despite genotype-specific differences in sleep/wake architecture and in extracellular adenosine dynamics, we did not observe changes in cortical neural dynamics (Fig 8E and 8F). Thus, it remains unclear how astrocytic H1R modulation of extracellular adenosine shapes surrounding neuronal dynamics in the cortex. One caveat of the sleep/wake experiments presented here is that EEG recordings may lack the spatial resolution to accurately track the neuronal changes mediated by shifts in extracellular adenosine, which were measured using fiber photometry, a technique that captures more spatially localized changes than EEG. Thus, deletion of astrocytic H1R may alter more local neuronal dynamics across sleep/wake than detected here via EEG. In light of recent work demonstrating that astrocyte-specific adrenergic signaling shapes synaptic weights via activation of neuronal adenosine receptors [38], along with our results demonstrating that astrocytic H1R shapes NE-induced astrocyte $Ca^{2+}$ and extracellular adenosine levels, one line of investigation would be to test whether H1R contributes to NE-mediated synaptic modulation important for memory consolidation occurring across sleep/wake.

Ex vivo electrophysiological experiments in WT and H1R cKO slices could illuminate in detail how astrocytic H1R signaling modulates adenosine-dependent changes in synaptic activity, while more local in vivo recording techniques—such as optrodes—could reveal the correlations between adenosine fluctuations, or astrocyte $Ca^{2+}$ fluctuations, and state-specific neuronal dynamics and behavior.

## Conclusions

By demonstrating that a single astrocyte-specific GPCR—H1R—shapes astrocyte $Ca^{2+}$ dynamics, extracellular adenosine flux, and sleep/wake behavior, our results contribute to an emerging model in which astrocytes act as a central hub for neuromodulator sensing and arousal regulation. In contrast to the previously described wake-promoting effect of H1R, our results indicate that astrocytic H1R in cortex reduces wakefulness and promotes REM sleep, highlighting the cell-type specificity of histaminergic arousal control. Glial-specific neuromodulatory GPCRs may therefore be a new therapeutic target for neurological or neuropsychiatric disorders in which sleep is disrupted. To leverage this possibility, we will need to further define how distinct neuromodulators shape astrocyte physiology on minutes-long timescales, detail the signal transduction pathways that underlie astrocytic control of extracellular adenosine levels, and demonstrate the downstream effects on cortical synaptic activity across sleep/wake.

## Methods

### Ethics statement

All experimental protocols were conducted according to the National Institutes of Health (NIH) guidelines for animal research and approved by the Institutional Animal Care and Use Committee (IACUC) at University of California, San Francisco (IACUC protocol AN203400).

### Animals

Mice were provided food and water a*d libitum* while being housed on 12:12 light-dark cycle. Transgenic floxed-H1R mice (H1R<sup>fl/fl</sup>) were generated by GemPharmatech (strain number T007142) using the CRISPR/Cas9 system to insert loxP sites flanking exon 3, which contains most of the H1R coding region. In vivo experiments were carried out in adult WT C57BL/6J mice and transgenic H1R<sup>fl/fl</sup> mice at 8–12 weeks old. Ex vivo experiments were done in P28–58 Swiss, C57BL6, or H1R<sup>fl/fl</sup> mice. Male and female mice were used in experiments as available. All sleep recordings for fiber photometry and electrophysiology experiments were performed during the light cycle.

### Ex vivo methods

**Surgical procedures.** The fluorescent $Ca^{2+}$ indicators GCaMP6f or jRGECO were virally expressed via neonatal, intracortical virus injections. Pups (P0–3) were anesthetized on ice for 3 min. A glass needle with ~5 µm diameter tip was loaded with 2 µl of either *AAV5-GfaABC1D-cyto-GCaMP6f-SV40* (1.9E13 vg/mL; custom virus from University of Pennsylvania Vector Core using Addgene plasmid #52925) or *AAV9-pGp-GfaABC1D-jRGECO1b* (3.19E13 vg/mL; custom virus from Vigene Biosciences) and 0.5 µl of Fast Green dye. After positioning pups on a digital stereotax, coordinates were zeroed at lambda. Virus was then injected at four injection sites in a 2 × 2 square grid. The first injection site was 0.8–0.9 mm lateral and 1.5–1.6 mm rostral from lambda and all four injection sites were 0.8 mm apart. Using a microsyringe pump (UMP-3, World Precision Instruments), 30–80 nl of virus was injected at two depths (0.1 and 0.2 mm below the skull) at each injection site at 3 nl/s. After recovery from anesthesia, all injected pups were returned to their home cage. For H1R cKO experiments, transgenic H1R<sup>fl/fl</sup> pups were injected with 60–80 nl of *AAV5-GFAP(0.7)-RFP-T2A-iCre* (5.0E12 vg/mL; Vector Biolabs, VB1133) and *AAV5-GfaABC1D-cyto-GCaMP6f-SV40* at a 1:1 ratio. The same procedure was used for experiments using *AAV9-pGp-GfaABC1D-jRGECO1b* to measure astrocyte $Ca^{2+}$, however, *AAV5-GFAP(0.7)-EGFP-T2A-iCre* (2.69E13 vg/mL; Vector Biolabs, VB1131) was used instead of the RFP-tagged Cre virus.

**2P imaging.** Coronal, acute V1 slices (300-µm thick) were collected from mice P28–58. For acute slice preparation, mice were decapitated according to an IACUC-approved protocol and the brain was dissected while submerged in ice-cold sucrose solution. Slices were cut in ice-cold sucrose solution using a vibratome (VT 1,200, Leica) and immediately transferred to room temperature (RT), continuously aerated (95% $O_2$/5% $CO_2$) standard artificial cerebrospinal fluid (ACSF). After collection, all slices were incubated in a 37°C water bath for 30 min and then kept at RT for imaging. For bath application experiments, each slice was allowed to rest for at least 5 min in the recording chamber with recirculating ACSF and 1 µM TTX to block neuronal action potentials. An imaging plane, at an average slice depth of ~45 µm, was selected based on reporter expression density and astrocyte health. Images were acquired using a custom-built 2P microscope that included an upright fluorescence microscope and the basic components for 2P imaging: one Ti:Sa laser (MaiTai, SpectraPhysics), Pockel cell (Conoptics) for modulating laser power, linear galvanometer (Cambridge Technology) for laser scanning, emission filters, photomultiplier tubes (Hamamatsu), and ScanImage software (version SI5.7) for image acquisition. Laser power was set such that astrocyte somas were visible, but laser-induced astrocyte GCaMP6f or jRGECO activity was minimal.

**Image acquisition for dose-response experiments:** Dose-dependent GCaMP6f dynamics were captured using a 60×, 1.2 N.A. objective (Nikon), 920 nm excitation, and a 525/50 nm emission filter. Images were acquired using a 512 × 512 pixel field-of-view (FOV), 0.34 µm/pixel spatial resolution, and 1.42 Hz frame rate. Spontaneous GCaMP6f dynamics were recorded during a 1–2 min baseline prior to pipetting HA into the recirculating ACSF bath.

**Image acquisition for H1R pharmacology experiments:** H1R-dependent GCaMP6f dynamics were captured as described, but using a 20×, 1.0 N.A. objective (Olympus), for a spatial resolution of 1.04 µm/pixel.

**Image acquisition for H1R cKO experiments:** GCaMP6f or jRGECO dynamics were captured as described for pharmacology experiments. Cre-RFP fluorescence was imaged immediately after GCaMP6f imaging using 1,040 nm excitation and a 605/15 nm emission filter. For NE experiments on untreated slices, jRGECO dynamics were captured using 1,000 nm excitation at 1.42 Hz and Cre-GFP fluorescence was imaged immediately after jRGECO imaging using 920 nm excitation. Each video was 5–10 min total and spontaneous GCaMP6f or jRGECO dynamics were recorded during a 1–2 min baseline prior to pipetting HA or NE into the recirculating ACSF bath. For the sequential ligand experiment shown in Fig 5A, GCaMP6f dynamics were imaged for 9 min after 0.5 µM HA addition and then for another 1-min baseline prior to 10 µM NE addition.

**Pharmacology.** Details for pharmacological reagents used in ex vivo imaging experiments are as follows: HA (Sigma-Aldrich, H-7250) was dissolved in ACSF prior to each experiment and pipetted into a 10–20 mL recirculating ACSF bath for a final HA concentration of 0.5, 5, 50, or 100 µM. NE (Sigma-Aldrich, A7256) solution was pipetted into the ACSF bath for a final NE concentration of 10 µM. The H1R-specific antagonist chlorpheniramine (Sigma-Aldrich, C3025) was used at 50 µM and Tetrodotoxin Citrate (Hello Bio, HB1035) was used at 1 µM. Slices were incubated in inhibitors for >5 min prior to 2P imaging.

**2P fluorescence event detection.** GCaMP6f and jRGECO 2P videos were characterized using AQuA software [56] and custom MATLAB code. Before AQuA analysis, 2P videos were smoothed by averaging every 2–4 frames to reduce noise. For accurate detection of fluorescent events, signal detection thresholds were manually adjusted for each 2P video using AQuA's graphical user interface, keeping parameters similar within each experiment.

For GCaMP6f or jRGECO analysis in H1R cKO slices, WT and H1R cKO ROIs were manually drawn based on co-expression patterns of the tagged Cre and the $Ca^{2+}$ indicator, which were determined by overlaying GCaMP6f and RFP (or jRGECO and GFP) z-projections. To make an ROI map for AQuA detection, ROIs were overlaid and filled on a black background (512 × 512 pixels). This image was flattened and converted to an 8-bit image, which was saved as a tiff and loaded into AQuA for ROI-specific event detection.

**AQuA output analysis.** After event detection, AQuA results were quantified using custom MATLAB code for each experimental condition. The key analysis steps are described below.

**Dose-response data:** Percent of active pixels in Fig 1 was defined as number of AQuA detected pixels per frame relative to total pixels in the field of view. Baseline activity was defined as average % active pixels prior to HA addition. The baseline was subtracted from % active pixels to obtain Δ pixels active over time. For each slice, the maximum Δ pixels active was found, and the area under the Δ pixels active curve (AUC) was calculated across the 120 s period post-HA. Statistical differences between AUC or maximum Δ pixels active for each HA concentration were detected using MATLAB's one-sided Wilcoxon rank-sum test.

**Pharmacology data:** The Δ pixels active over time and maximum Δ pixels active in Fig 2 were calculated as above. Maximum Δ pixels active for no-antagonist and +H1R-antagonist conditions were compared using MATLAB's one-sided Wilcoxon rank-sum test. To analyze $Ca^{2+}$ event characteristics, events pre-HA (120 s period during baseline) and post-HA (120 s immediately following HA) were pooled from all slices. Mean and 95% confidence interval for event amplitude, area, and duration were estimated using bootstrapping (10,000 samples with replacement) for no-antagonist and +H1R-antagonist data. Statistically significant differences between pre- and post-HA conditions were detected via a permutation test (1,000 permutations) on pooled data before bootstrapping.

**H1R cKO $Ca^{2+}$ data:** The Δ pixels active over time and maximum Δ pixels active in Fig 4 were calculated as above with the following change. The % active pixels was calculated relative to the total number of pixels subsumed by either all WT or cKO ROIs, instead of total number of pixels in the FOV. MATLAB's one-sided Wilcoxon signed-rank test was used to detect statistically significant differences between WT and cKO maximum Δ pixels active post-HA or post-NE.

**Δ*F*/*F* analysis.** For data shown in Fig 5, WT and cKO ΔF/F was calculated as follows: in ImageJ, WT and cKO ROIs were overlaid on 2P videos that were smoothed by averaging every 2–4 frames to reduce noise. Mean gray values over time were then extracted for each ROI. Baseline fluorescence ($F_0$) was the median gray value across a 60 s baseline period prior to ligand addition, which was used to calculate ΔF/F [$(F − F_0)/F_0$] per ROI. To quantify response magnitudes, AUC was calculated via Python's np.trapz function across a 30 s pre-ligand period and a post-ligand period (60 s for 0.5 µM HA and 30 s for 10 µM NE).

## Validation of H1R cKO

Deletion of H1R transcripts in Cre-RFP+ astrocytes was verified using RNAscope and immunohistochemistry (IHC) as follows: Neonatal virus injections were done as described to drive Cre-RFP expression in cortical astrocytes. Mice (~P28) were deeply anesthetized with vaporized isoflurane (1%–1.5% vol.) and then intracardially perfused with ~10 mL of ice-cold 1× PBS and then ~10 mL of ice-cold 4% PFA. The brain was dissected out, fixed in 4% PFA overnight at 4°C, incubated in 30% sucrose at 4°C until no longer floating, and then embedded in OCT and stored at −80°C. A cryostat was then used to collect 14–20 µm sections from V1, which were stored at −80°C.

**H1R mRNA probe design.** Probe design was based on the sequence deletion in H1R cKO mice. ACDBio designed and generated an RNAscope probe that targeted bases 269–1,342 in the H1R gene sequence (GenBank accession #: NM_001410031.1).

**H1R mRNA, S100β, and Cre-RFP labelling.** Sections were thawed to RT and washed for 5 min in 1× PBS in a Coplin jar (container used for all washes unless specified). Slides were then baked for 15 min at 60°C in a dry air oven (used for all incubations above RT). Sections were dehydrated for 5 min each in RT serial dilutions of ethanol (EtOH): 50%, 70%, and 100%. Sections were air dried for 5 min and then incubated in 2–3 drops of ACDBio $H_2O_2$ for 10 min at RT. After two washes in de-ionized $H_2O$ (di$H_2O$), sections were washed in 100% EtOH and air dried. A pap pen was used to draw a hydrophobic barrier around each section. The barrier was allowed to dry, and sections were incubated for 15 min at 40°C in 2–3 drops (per section) of ACDBio protease III solution. Sections were washed three times in di$H_2O$ at RT. Then ~50 µl of ACDBio H1R mRNA probe was added to each section and hybridized for 2 h at 40°C. During hybridization the following reagents were brought to RT: ACDBio AMP1–3 solutions, HRP-C1, HRP blocker, and 1:1,000 690 Opal dye in ACDBio TSA buffer. After hybridization, sections went through the following incubation steps and between each step sections were

washed 3× in ~500 µl per section of ACDBio wash buffer: (1) 2–3 drops of AMP-1 for 30 min at 40°C; (2) 2–3 drops AMP-2 for 30 min at 40°C; (3) 2–3 drops AMP-3 for 15 min at 40°C; (4) 2–3 drops of HRP-C1 for 15 min at 40°C; (5) ~50 µL of 1:1,000 690 Opal dye for 30 min at 40°C; (6) 2–3 drops HRP blocker for 15′ at 40°C. Next, sections were processed for IHC. Sections were permeabilized and blocked by incubating for 60 min at RT in ~200 µl of antibody incubation solution (AIS: 1× PBS with 0.05% Triton X-100, 3% fetal bovine serum, 0.025% sodium azide). AIS was removed and sections were incubated overnight at 4°C in 50–75 µL of primary antibody mixtures diluted in AIS: 1:250 rabbit α S100β (Sigma-Aldrich, SA-85500172), 1:1,000 rat α mCherry (Invitrogen, M11217). Sections were washed 3× in 1× PBS, 10 min each at RT. Then, sections were incubated for 2 h at RT in 50–75 µl of secondary antibody mixtures diluted in AIS: 1:2,000 goat α rabbit 488 (Invitrogen, A11034), 1:2,000 goat α rat 555 (Invitrogen, A21434). Sections were washed 3× in 1× PBS, 10 min each at RT and then incubated in 2–3 drops of ACDBio DAPI for 30 s at RT. Slides were gently tapped to remove DAPI, and Fluoromount (SouthernBiotech, 0100-01) was pipetted onto each section. Finally, a coverslip was added to each slide and labelled sections were stored at 4°C until imaging.

**RNAscope and IHC image collection and analysis.** A Zeiss LSM 880 point-scanning confocal microscope was used to image RNAscope and IHC labelling. Z-stacks approximately spanning each section were collected using a 63×, 1.4 N.A. oil immersion objective. All images had a 512 × 512 pixels FOV and 16-bit depth. H1R mRNA, RFP, S100β, and DAPI were imaged using 633, 561, 488, and 405 nm lasers, respectively. For each section, 1–3 FOVs were imaged across V1 layers. After image collection, laser channels were separated, and ROIs were drawn around easily identified astrocyte soma based on RFP and S100β signal. ROIs were then overlaid on H1R mRNA images (633 nm channel), and H1R puncta within ROIs were manually counted based on colocalization with S100β and RFP signal.

**IHC and colocalization analysis of NeuN/Cre-RFP.** To verify lack of neuronal Cre-RFP expression, we used the above IHC procedure to co-label neurons and RFP expression in sections from H1R$^{fl/fl}$ mice that were neonatally injected with *AAV5-GFAP(0.7)-RFP-T2A-iCre,* as described above*.* For primary antibody labelling, we used 1:1,000 rat α mCherry (Invitrogen, M11217) and 1:1,000 rabbit α NeuN (EMD Millipore Corp, ABN78) to label RFP expressing cells and neurons, respectively. For secondary antibody labelling, we used 1:2,000 goat α rabbit 488 (Invitrogen, A11034), 1:2,000 goat α rat 555 (Invitrogen, A21434). After mounting, z-stacks were collected as described above using a 20×, 0.8 N.A. air immersion objective. RFP, NeuN, and DAPI were imaged using 561, 488, and 405 nm lasers, respectively. For image analysis, laser channels were separated, and ROIs were manually drawn around RFP+ soma. In ImageJ, the RFP and NeuN channels were smoothed using the Gaussian blur filter (sigma radius = 1.00) and then binarized. Somatic ROIs were overlaid on each binarized image and the % of NeuN and RFP+ positive pixels (relative to total ROI pixels) was calculated for each ROI.

### *In vivo* methods

**Surgical procedures.** One to two hours before surgery, mice were administered dexamethasone (5 mg/kg, s.c.). At the time of surgery, mice were anesthetized with vaporized isoflurane (1%–1.5% vol.) and administered 0.05 mg/ kg buprenorphine (s.c.) and 5 mg/kg carprofen (i.p.). Mice were positioned on a digital stereotax and lidocaine was administered over the surgery site (−2.5 mm lateral, +0.5 mm rostral from lambda). Following surgery, all animals were singly housed and given additional enrichment and post-operative care with close monitoring.

*Fiber photometry + EEG/EMG surgery* (Figs 6, 7, S4, S6, and S7): Following anesthesia and pain management procedures described above, a craniotomy window was opened over left V1 and we subsequently injected −0.2 mm below cortex with 1,000 nl viral vector mixtures at 2 nl/s. To measure WT astrocyte Ca$^{2+}$ and extracellular histamine or adenosine dynamics, C57BL6 mice were co-injected with either *AAV9-hSyn-GRAB-Adenosine1.0* (1.0E13 vg/mL; premade virus from WZ Biosciences) or *AAV9-hsyn-GRABhis1.0* (8.81 E13 vg/mL; premade virus from WZ Biosciences) and *AAV9-pGp-GfaABC1D-jRGECO1b* (same virus used for slice experiments) at a 2:1 ratio. For measuring astrocyte Ca$^{2+}$ dynamics in H1R cKO mice, H1R$^{fl/fl}$ mice were co-injected with viral vectors *AAV5-GFAP(0.7)-EGFP-T2A-iCre* (same virus used

for slice experiments) and *AAV9-pGp-GfaABC1D-jRGECO1b* at a 2:1 ratio. To measure astrocyte Ca$^{2+}$ and extracellular adenosine in H1R cKO mice, H1R$^{fl/fl}$ mice were co-injected with viral vectors *AAV5-GFAP(0.7)-T2A-iCre* (7.8E12 vg/mL; Vector Biolabs, VB4887), *AAV9-hSyn-GRAB-Adenosine1.0, and AAV9-pGp-GfaABC1D-jRGECO1b* at a 2:2:1 ratio. For all mice, a fiber optic cannula (Mono Fiberoptic Cannula, 400-µm core, 430 nm, 0.66 NA, 1 mm length, Doric Lenses) was then lowered to the depth of viral injections. To track sleep/wake in these mice, a screw electrode (8403, Pinnacle) was implanted in right V1 (−2.5 mm lateral, +0.5 mm rostral from lambda), and soldered onto a biosensor head mount (8402, Pinnacle). A reference electrode was placed in the contralateral frontal cortex (−1.25 mm lateral, +2.70 mm rostral from bregma) and EMG probes from the head mount were inserted into the trapezius muscles. The head mount and fiber optic cannula were secured in place using dental cement (C&B Metabond, Parkell).

***EEG/EMG surgery for WT versus cKO sleep/wake analysis (Fig 8):*** To measure sleep/wake dynamics from the cortical hemisphere ipsilateral to Cre- or sham virus transduction, a craniotomy window was opened over right V1 in H1R$^{fl/fl}$ mice and subsequently injected at −0.2 mm with 1,000 nl of *AAV5-GFAP(0.7)-EGFP* (1.3E13 vg/mL; Vector Biolabs, VB1149) or *AAV5-GFAP(0.7)-EGFP-T2A-iCre*, at 2 nl/s. A screw electrode was implanted directly over the virus injection site and soldered onto a biosensor head mount. A reference electrode and EMG wire were implanted, and the head mount was secured as described above.

**Fiber photometry and electrophysiology recordings.** Animals were given 2 weeks post-surgery for recovery and viral expression. They were then habituated in a freely moving recording setup for at least one 2-h session on the day or two before experimental recordings began. The recording chamber was a plexiglass cylinder (height = 8 in; diameter = 10 in) filled with bedding from the animal's home cage, which was cleaned with 70% ethanol between habituation and recording sessions. For each experimental recording, the chamber contained fresh bedding material and the mouse's nest, which was transferred from its home cage. 1–2 recordings were done per day, with the first starting ~7 ZT and the second ~10 ZT (mean recording time of day rounded to the nearest 0.5 hour was 9 ZT with a range of 6.5–11.5 ZT). On recording days, mice (Figs 6–8, S4, S6, and S7) were plugged in and recordings were started immediately. All cables tethered to the animal's head mount were run through a rotary joint (Doric Lenses), which was attached to a bar secured to the top of the recording chamber. Experimental recordings lasted >195 min (final recording lengths were shorter after signal processing and noise clipping) and each animal underwent recordings at least 1–2 times per week until 5-weeks post-surgery when animals were euthanized and perfused for IHC to validate implant locations and viral expression.

Fiber photometry was performed on a Tucker-Davis Technologies RZ10X processor (TDT) outfitted with the Lux photometry module (integrated 405, 465, and 560 nm LEDs; photosensors; low-autofluorescence 400 µm, 0.57 NA patch cords; Doric Lenses). All three LEDs were driven concurrently, but each was sinusoidally modulated at a unique carrier frequency (lock-in multiplexing), allowing deconvolution of the 405 nm "control," 465 nm (jRGECO), and 560 nm (GRAB-Ado) channels at the photoreceiver. Demodulated emission signals were digitized at 1,017 Hz and low-pass filtered at 6 Hz in TDT Synapse software and collected through bandpass filters: 500–540 nm for 405/465 nm excitation, and 580–680 nm for 560 nm excitation. Simultaneously, EEG and EMG data were collected using a 1,017 Hz sampling rate with a 30 Hz low-pass filter through a three-channel Pinnacle EEG/EMG Data Acquisition System connected to the TDT RZ10X processor.

**Data analysis.** Custom code written in Python (3.9.7) was used for signal processing and analyzing fiber photometry and electrophysiology data acquired through the Tucker-Davis Technologies (TDT) RZ10X Processor. Core packages include tdt (0.5.0) to read in the TDT RZ10X data, pandas (1.3.4) for data wrangling and creating data structures for analysis, numpy (1.20.3) and scipy (1.7.1) for mathematical and statistical operations, and matplotlib (3.4.3) and seaborn (0.11.2) for data visualization.

**Fiber photometry signal processing.** Fiber photometry signal data was low-pass filtered with a 3 Hz cutoff. Data in the 465 and 560 nm channels were then smoothed using a Savitzky-Golay (SG) filter with a 5 s window and detrended using Python's signal.detrend function to account for photobleaching. The 405 nm (isosbestic) channel was further

low-pass filtered using scipy.signal.firwin with a 1 Hz cutoff, smoothed using the SG filter, and fitted to the 465 and 560 nm channels through ordinary least squares regression to remove motion and hemodynamic artifacts. Then, 465 and 560 nm signals were z-scored using Python's stats.zscore function. Finally, for all state transition-triggered averages (Figs 6 and 7), the z-scored photometry signal was normalized by calculating the absolute change relative to the signal mean during the pre-transition period, which was 200 s for NREM/wake transitions, 30–60 s for REM transitions.

**EEG/EMG signal processing.** EEG signals were detrended using a reverse exponential fit and large noise artifacts were deleted by z-scoring and clipping EEG signals >8 standard deviations. Then, manual inspection of EEG and EMG time frequency spectrograms was performed as quality control. When possible, easily identified noise artifacts were cut out from recordings. If EEG or EMG recordings were excessively contaminated with noise, the electrophysiology and accompanying fiber photometry recording were removed from final datasets.

**Sleep/wake analysis.** For sleep scoring, multitaper spectral analysis (number of tapers = 29; window = 30 s; step size = 1 s) was performed on EEG data [92] to determine average delta (0.5–4 Hz) power over time. These data were used to train a HMM (hmmlearn, GitHub) to infer NREM and wake states in each recording. For data in Fig 8, EMG gamma power (65–100 Hz), which was calculated with the same multitaper spectral analysis described above, was also used to infer NREM and wake bouts. For animals with photometry (Figs 6, 7, S4, S6, and S7), EMG data was excluded from the HMM due to noise artifacts in photometry recordings, which could be interpreted by a human experimenter but not by the automated model. In the HMM model, wake and NREM periods were scored with 1 and 0, respectively. To identify behavioral states with high confidence, 100 iterations of the HMM were averaged. NREM and wake bouts were then identified based on HMM scores across the EEG recording (NREM < 0.45; wake > 0.55) and minimum bout durations (NREM ≥ 100 s and wake ≥ 30 s). HMM-predicted states were plotted alongside z-scored EEG theta/delta ratio and z-scored EMG gamma power to facilitate manual REM bout detection. Then, HMM-detected wake bouts were manually reclassified as REM if they (1) followed NREM period, (2) exhibited a theta/delta peak > median + 2 standard deviations, and (3) showed minimal to no EMG gamma power. REM bouts—defined by a theta/delta peak with minimal EMG gamma—were often followed by a brief increase in EMG gamma power, consistent with brief awakenings after REM sleep in mice. These were manually scored as wake. We note that wake bouts were not detected after every REM bout, most likely due to the seconds-level temporal resolution of our multitaper analysis, which could smooth out brief elevations in EMG gamma, leading to detection of fewer brief awakenings. By using a 30 s window with a 1 s step, we sacrificed temporal resolution for frequency resolution for accurate HMM detection of state transitions that span tens of seconds to minutes. Using this method, 91%–96% (Figs 8 and S7) of each recording was classified as wake, NREM, or REM, leaving 4%–9% unclassified.

For state-dependent power spectral EEG analysis, multitaper spectrograms (generated using parameters above) were separated into wake, NREM, or REM periods. PSDs for each state were then calculated by averaging power in the 1–20 Hz range for concatenated wake, NREM, and REM spectrogram periods. PSDs were normalized by total power in each state and displayed as a percentage. Relative band power in each state was similarly calculated for delta (1–4 Hz), theta (6–10 Hz), and sigma (10–15 Hz) frequency bands from the multitaper spectrograms. Band powers were normalized to total power in each behavioral state to give % power for wake, NREM, and REM.

**Histology and immunohistochemistry.** Mice were deeply anesthetized with vaporized isoflurane (1%–1.5% vol.) and intracardially perfused with ~10 mL of ice-cold 1× PBS and then ~10 mL of ice-cold 4% PFA. The brain was carefully dissected out to preserve photometry fiber and EEG screw tracks and then fixed in 4% PFA overnight at 4°C, incubated in 30% sucrose at 4°C until no longer floating, frozen on dry ice and stored at −80°C. For slicing, the brain was brought to −20°C, embedded in OCT, and then coronally sectioned (40 μm) on a cryostat.

For antibody staining, sections were first washed 3× in 1× PBS and then permeabilized at R.T. in 0.01% TritonX in PBS for 30 min. Sections were blocked in 10% NGS (Invitrogen) for 1 h at RT and then incubated O/N at 4°C in primary antibodies diluted in 2% NGS: 1:3,000 chicken α GFP (Abcam, ab13970) for labelling Cre-GFP, GRAB-HA or GRAB-Ado and

1:2,000 rat α mCherry (Invitrogen, M11217) for labelling jRGECO. Next, sections were wash 3× in 1× PBS before 2-h R.T. incubation in secondary antibodies diluted in 2% NGS: 1:2,000 goat α chicken 488 (Invitrogen, A11039) ± 1:2,000 goat α rat 555 (Invitrogen, A21434). Finally, sections were washed 3× in 1× PBS, mounted in Fluoromount with DAPI, and cover-slipped. Sections were imaged using confocal microscopy.

## Supporting information

**S1 Fig. Dose-dependence of histamine (HA)-triggered features in astrocyte Ca$^{2+}$. (A)** HA-triggered change in Ca$^{2+}$ event rate (10-s time bins) relative to mean 60 s pre-HA for 0.5, 5, and 100 μM HA, indicated by red, green, and blue traces, respectively. Traces = mean across slices. Shaded error bars = SEM. Purple bar = HA time in recirculating ACSF. **(B)** Summary of data in A. Maximum change in event rate during 2 min post-HA for each slice, with mean ± SEM at right. *p*-values via one-sided Wilcoxon rank-sum test. **(C)** Distribution of event amplitude, area, and duration in 60-s time bins pre- and post-HA addition for 0.5, 5, and 100 μM HA. Each box spans interquartile range (25th–75th percentile), with horizontal line indicating median, whiskers extending to most extreme values within 1.5× the IQR, and outliers plotted individually. *p*-values via one-sided Wilcoxon rank-sum test.
(TIF)

**S2 Fig. Two somatic regions of interest (ROIs) that exhibit substantial Cre-RFP and NeuN overlap exhibit RFP$^+$ and NeuN$^+$ pixels in distinct z-planes. (A)** Percentage of Cre-RFP$^+$ astrocyte soma pixels that are NeuN$^+$. Arrows point to two cells with >50% RFP and NeuN overlap, which was quantified by generating summed z-projection of NeuN channel, binarizing the z-projection, and then calculating the % of NeuN$^+$ pixels in somatic ROIs. **(B–C) Left:** Traces show fluorescence level of DAPI (blue), NeuN (green), RFP (pink) across all z-planes in z-stack for ROI 1 and 2 indicated in panel A. Vertical lines show z-plane of maximum fluorescence for NeuN (green) and RFP (pink), revealing no overlap. **Right:** Confocal micrographs show RFP and NeuN fluorescence in ROI 1 indicated by yellow arrows and ROI 2 indicated by cyan arrows. Top rows: RFP fluorescence and lack of NeuN fluorescence in z-plane displaying maximum RFP fluorescence. Bottom rows: RFP and NeuN fluorescence in z-plane displaying maximum NeuN fluorescence. **(B)** ROI 1 displays absence of RFP in NeuN max z-plane. **(C)** ROI 2 exhibits reduced RFP levels and captures the edges of multiple NeuN$^+$ cells rather than an RFP$^+$ individual neuron. Scale bars = 20 μm.
(TIF)

**S3 Fig. Histamine-1-receptor (H1R) deletion in astrocytes leads to reduced cell area subsumed by histamine (HA)-triggered Ca$^{2+}$ while leaving wild-type (WT) levels of event synchrony intact. (A)** Example z-projection of H1R cKO astrocytes identified via live 2P RFP imaging. H1R cKO astrocytes outlined in yellow. WT astrocytes were identified based on GCaMP6f expression (not shown) and are outlined in white. Scale bar = 50 μm. **(B)** Sample z-projection of 2P image showing pre- and post-HA AQuA events. WT cells are filled by AQuA events post-HA, while cKO regions of interest (ROIs) exhibit an increased number of discrete AQuA events relative to pre-HA. Pre- and post-HA z-projections include the same number of frames. Scale bars = 50 μm. **(C)** Mean percent-ROI-active during 1 min pre-HA and 2 min immediately following HA. WT ROIs exhibit increased activity (post – pre = 19.9%). cKO cells show a small increase (post – pre = 2.3%). For C–E, data shown are mean ± 95% CI. **(D)** Ca$^{2+}$ event area per cell during 1 min pre-HA and 30 s immediately following HA. WT astrocytes exhibit larger Ca$^{2+}$ events (post – pre = 59.7 μm$^2$) post-HA, while cKO event area shows no change. Post-HA window is 30 s because area increase is short-lived. **(E)** Events/min per cell during 1 min pre-HA and 2 min immediately following HA. WT and cKO astrocytes exhibit increased event rate post-HA (WT post – pre = 10.4 events; cKO post – pre = 4.3 events). For C–E, means and 95% CI are estimated via bootstrapping with replacement; *p*-values are calculated via permutation test on data before bootstrapping. Data in panels C–E collected from 9 slices and 3 mice.
(TIF)

**S4 Fig. Extracellular histamine (HA) recordings confirm previous sleep- and wake-specific findings. (A)** Experimental schematic showing V1 GRAB-HA virus injection with photometry fiber and EEG screw placement in c57bl6 mice. $N = 6$ recordings across 3 mice. **(B)** Schematic showing freely moving fiber photometry and electrophysiology recording setup. **(C)** Example z-scored GRAB-HA photometry trace aligned to color-coded wake (yellow), NREM (blue), and REM (purple) periods. **(D–F)** Event-triggered averages of change in GRAB-HA fluorescence relative to mean of pre-transition period. Trace = mean GRAB-HA activity. Shaded error bar = SEM. **(D)** NREM-to-wake transitions spanning 560 s. Trace shows mean of 27 transitions. **(E)** Wake-to-NREM transitions spanning 560 s. Trace shows mean of 40 transitions **(F)** NREM-to-REM transitions spanning 180 s. Trace shows mean of 8 transitions. Note: too few REM-to-wake transitions (3 total) identified for quantification of mean GRAB-HA levels.
(TIF)

**S5 Fig. Localization of astrocyte-specific virus transduction, EEG screw, and photometry fiber in H1R<sup>fl/fl</sup> brains.**
**(A)** Representative confocal micrograph of histamine-1-receptor (H1R) cKO brain slice for fiber photometry datasets. Astrocytic Cre-GFP (green) and jRGECO (magenta) is expressed below photometry fiber track in right hemisphere. EEG screw track visible in contralateral hemisphere. Scale bar = 500 μm. **(B)** Left: confocal micrograph showing DAPI (yellow), Cre-GFP (green), jRGECO (magenta), and S100β (blue) expression below photometry fiber track shown in A. White arrow indicates GFP⁺/jRGECO⁺/S100β⁺ astrocyte. Scale bar = 100 μm. Right: separated confocal channels showing DAPI, GFP, jRGECO, and S100β expression in outlined astrocyte indicated with white arrow in left image. Scale bars = 10 μm. **(C–D)** Representative confocal micrographs of sham (C) and cKO (D) brain slices for dataset shown in Fig 8. Astrocytic Cre-GFP or GFP (green) and GFAP (magenta) expression below EEG screw track in right hemispheres. White boxes outline areas shown in E–H. Scale bars = 500 μm. **(E–H)** Confocal micrographs in E, F and G, H show GFP⁺ (green) and GFAP (magenta) expression in outlined regions in C and D, respectively. **(E, G)** Confocal micrograph showing astrocytic Cre-GFP expression in cortex and hippocampus below EEG screw track. White arrows indicate GFP⁺ astrocyte shown at a higher magnification in F and H. White dotted line = tissue edge of EEG screw track. Scale bars = 100 μm. **(F, H)** Higher magnification confocal micrograph showing GFP⁺ (green) and GFAP (magenta) co-expression in cortical astrocytes below EEG screw track. White arrows in F and H indicate GFP⁺/GFAP⁺ astrocyte in cortex shown by white arrows in E and G, respectively. Scale bars = 50 μm.
(TIF)

**S6 Fig. Sleep/wake characteristics in wild-type and histamine-1-receptor (H1R) conditional knockout (cKO) mice with EEG screw contralateral to photometry fiber and Cre-virus. (A)** Mouse surgery schematics show location of EEG screw, photometry fiber, and virus injections in WT and H1R cKO mice. Photometry fiber was implanted over V1 injection site for astrocytic jRGECO virus ± extracellular GRAB-Ado and astrocytic Cre or sham virus. EEG screw was implanted in contralateral V1. WT cohort included c57bl6 and H1R<sup>fl/fl</sup> mice. Color code (cKO in cyan, WT in gray) used in panels B–E. **(B)** Power spectral density (PSD) plots show WT (gray) and cKO (cyan) % power relative to total power across 1–20 Hz during wake, NREM, and REM. PSDs calculated via multitaper spectrogram. Traces = mean across mice; shaded error bar = SEM. **(C)** WT (left) and cKO (right) relative band power (% of total power in each state) for delta (1–4 Hz), theta (6–10 Hz), and sigma (10–15 Hz) during wake (yellow), NREM (blue), and REM (purple). Bars show overall mean and overlaid data points show mean per mouse. Error bars = SEM. *p*-values via Wilcoxon rank-sum test with Benjamini–Hochberg correction for multiple comparisons. **(D)** Event-triggered averages show change relative to mean of pre-transition period in theta power around NREM-wake (yellow) and NREM-REM (purple) transitions spanning 120 s for WT (left) and cKO (right). Traces = mean across mice; shaded error bars = 95% CI. **(E)** Summary statistics for data in D show increased theta power at REM-onset relative to wake-onset for WT (left) and cKO (right). Scattered data points show mean (per recording) % theta power during 60 s post transition for NREM-to-wake (yellow) and NREM-to-REM (purple). Data points with error bars = overall mean ± SEM. *p*-values via one-sided Wilcoxon rank-sum test.
(TIF)

**S7 Fig. Sleep/wake architecture in wild-type and histamine-1-receptor (H1R) conditional knockout (cKO) mice with EEG screw contralateral to photometry fiber and Cre-virus. (A)** % of individual recordings scored as wake, NREM, or REM. For A–D, bars show overall mean and overlaid data points show mean per mouse ± SEM (WT: 90.7 ± 0.8%, cKOs: 92.2 ± 1.0%). *p*-value calculated via one-sided Wilcoxon rank-sum test. **(B)** Mean % time (relative to total time scored) in wake, NREM, and REM. % time per state for WT: 42.8 ± 3.4 wake, 53.3 ± 2.7 NREM, 4.0 ± 0.8 REM; cKO: 40.1 ± 1.7 wake, 56.2 ± 1.5 NREM, 3.6 ± 0.7 REM. One-sided Wilcoxon rank-sum test detects no difference in % time in sleep/wake. **(C)** Mean bouts per hour for wake, NREM, and REM. Bouts per hour for WT: 4.7 ± 0.4 wake, 5.4 ± 0.4 NREM, 1.4 ± 0.3 REM; cKO: 3.1 ± 0.4 wake, 5.3 ± 0.3 NREM, 1.5 ± 0.3 REM. *p*-value via one-sided Wilcoxon rank-sum test. **(D)** Mean bout duration in minutes for wake, NREM, and REM. Mean bout duration (min) for WT: 5.7 ± 0.5 wake, 6.1 ± 0.4 NREM, 1.8 ± 0.1 REM; cKO: 8.38 ± 0.9 wake, 6.5 ± 0.3 NREM, 1.4 ± 0.1 REM. (TIF)

**S1 Video. Astrocyte Ca²⁺ response to 0.5 µM HA.** Example video showing 6 min of astrocytic GCaMP6f activity with AQuA overlay in response to 0.5 µM HA. Time of HA entry is indicated in top left corner. Display frame rate = 0.47 Hz. Scale bar = 20 µm. Data quantified in Figs 1 and S1. (AVI)

**S2 Video. Astrocyte Ca²⁺ response to 100 µM HA.** Example video showing 6 min of astrocytic GCaMP6f activity with AQuA overlay in response to 100 µM HA. Time of HA entry is indicated in top left corner. Display frame rate = 0.47 Hz. Scale bar = 20 µm. Data quantified in Figs 1 and S1. (AVI)

**S3 Video. HA-triggered Ca²⁺ in wild-type (WT) and histamine-1-receptor (H1R) cKO astrocytes.** Example video showing 10 min of GCaMP6f activity with AQuA overlay in WT (white regions of interest [ROIs]) and H1R cKO (yellow ROIs) astrocytes. Time of 50 µM HA entry is indicated in top left corner. Display frame rate = 0.71 Hz. Scale bar = 50 µm. Data quantified in Figs 4B, 4D and S3C–S3E. (AVI)

**S4 Video. NE-triggered Ca²⁺ in wild-type (WT) and histamine-1-receptor (H1R) cKO astrocytes.** Example video showing 7 min of jRGECO activity with AQuA overlay in WT (white regions of interest [ROIs]) and H1R cKO (yellow ROIs) astrocytes. Time of 10 µM NE entry is indicated in top left corner. Display frame rate = 0.71 Hz. Scale bar = 50 µm. Data quantified in Figs 4C, 4F, 5F, and 5G. (AVI)

## Acknowledgments

We thank members of the Poskanzer lab for helpful discussion throughout this study: Nicole Werner for assistance with animal husbandry, Alba Peinado for building the 2P microscope and providing essential technical support, Caitlin Durkee for building and troubleshooting the fiber photometry and electrophysiology recording set-up, and Jennifer Thompson for administrative support. We thank Catherine Morgans for the RNAscope/immunohistochemistry staining protocol and Marci Rosenberg for technical advice.

## Author contributions

**Conceptualization:** Charlotte R. Taylor, Trisha V. Vaidyanathan, Kira E. Poskanzer.

**Formal analysis:** Charlotte R. Taylor, Vincent Tse, Drew D. Willoughby, Maxine Levesque.

**Funding acquisition:** Kira E. Poskanzer.

**Investigation:** Charlotte R. Taylor, Vincent Tse, Drew D. Willoughby, Kira E. Poskanzer.

**Methodology:** Charlotte R. Taylor, Vincent Tse, Drew D. Willoughby, Maxine Levesque, Kira E. Poskanzer.

**Project administration:** Charlotte R. Taylor, Kira E. Poskanzer.

**Resources:** Kira E. Poskanzer.

**Supervision:** Charlotte R. Taylor, Jeanne T. Paz, Kira E. Poskanzer.

**Visualization:** Charlotte R. Taylor.

**Writing – original draft:** Charlotte R. Taylor, Kira E. Poskanzer.

**Writing – review & editing:** Charlotte R. Taylor, Jeanne T. Paz, Kira E. Poskanzer.

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
