## [Editor Report · Decision Letter 0]

4 Feb 2025

Dear Dr Poskanzer, 

Thank you for submitting your manuscript entitled "Histamine-1-receptors regulate cortical astrocyte calcium and extracellular adenosine dynamics during REM sleep" for consideration as a Research Article by PLOS Biology.

Your manuscript has now been evaluated by the PLOS Biology editorial staff as well as by an academic editor with relevant expertise and I am writing to let you know that we would like to send your submission out for external peer review.

Once your full submission is complete, your paper will undergo a series of checks in preparation for peer review. After your manuscript has passed the checks it will be sent out for review. To provide the metadata for your submission, please Login to Editorial Manager (https://www.editorialmanager.com/pbiology) within two working days, i.e. by Feb 06 2025 11:59PM.

Kind regards,

Luke

Lucas Smith, Ph.D.

Senior Editor

PLOS Biology

lsmith@plos.org

---

## [Decision Letter · Decision Letter 1]

5 Mar 2025

Dear Dr Poskanzer,

Thank you for your patience while your manuscript "Histamine-1-receptors regulate cortical astrocyte calcium and extracellular adenosine dynamics during REM sleep" was peer-reviewed at PLOS Biology. It has now been evaluated by the PLOS Biology editors, an Academic Editor with relevant expertise, and by several independent reviewers. 

In light of the reviews, which you will find at the end of this email, we would like to invite you to revise the work to thoroughly address the reviewers' reports.

As you will see below, the reviewers find the study interesting and generally well done. However, they have also raised a number of important and sometimes overlapping concerns that we think will need to be thoroughly addressed before we can consider your paper for publication. The reviewers highlight that there may be technical issues with the sleep analyses performed here, and we think that additional validation of these results, with another algorithm and manual curation, will be required to bolster the conclusions. We also think that the manuscript should carefully address the reviewer concerns about possible hemodynamic artifacts, and the decision to conduct experiments in the light period, and we think the revision will need to provide the relevant clarifications and justifications requested by the reviewers. 

Given the extent of revision needed, we cannot make a decision about publication until we have seen the revised manuscript and your response to the reviewers' comments. Your revised manuscript is likely to be sent for further evaluation by all or a subset of the reviewers.

**IMPORTANT - SUBMITTING YOUR REVISION**

*Re-submission Checklist*

*Published Peer Review*

*PLOS Data Policy*

*Blot and Gel Data Policy*

Sincerely,

Luke

Lucas Smith, Ph.D.

Senior Editor

PLOS Biology

lsmith@plos.org

REVIEWS:

Reviewer #1: Taylor et al. present an interesting, well conducted and well presented study. They demonstrate direct effects of histamine on astrocytic calcium signaling through H1R and further show that astrocytic H1R deletion causes changes to astrocyte calcium responses in REM sleep as well as influencing adenosine levels in sleep. I have a few comments/suggestions:

(p 9, results). The authors show by measuring sleep with an EEG electrode in V1 in an area with localized knockdown of H1R that the knockout shifts the measured sleep/wake ratio. However, when measuring in the contralateral cortex to the viral injection no difference in sleep/wake ratio is found. In the discussion they propose that this observation may be a representation of local sleep. This is really an intriguing finding. However, the sleep/wake classification is based on EEG/EMG, and EMG in some cases are potentially missing (according to the methods section - not sure whether this comment relates to these observations). When examining Fig 6 spectrograms one gets the impression that there are subtle differences in particular in REM sleep, even though the aggregated statistics show no differences. 

1) With the automated classifier, with potential small differences in REM spectral components and lack of EMG in some recordings, how sure can the authors be that these differences in sleep/wake are not due to a mislabeling of for instance wake vs REM by their automated classifier? Were EMG recordings present in all the recordings in Fig 6 and Supplementary figure 5? This aspect could maybe be discussed, alternatively, could other thresholds or algorithms for sleep scoring be used to verify the findings? 

2)Were EEG recordings on ipsi- and contralateral side to the viral induced knockout of H1R recorded simultaneously in the same mouse, or in separate experiments? 

Sleep, and in particular REM sleep, is characterized by large hemodynamic changes that influence fluorescence recordings with relatively low SNR fluorescent sensors. 

3) If I understand the methods section correctly, all recordings of the GRABAdo have been corrected for with the "isosbestic signal" (405nm stimulation)? I'm not intimately familiar with these systems, but I presume there is a sequential activation of the LEDs of different colors? Some more details on this should be provided in methods. 

4) The absorption spectrum of hemoglobin varies quite significantly over the range light used (405/465/560). Can we be sure that this method adequately removes hemodynamic artifacts from the recordings? Or could even the correction introduce artifacts? Moreover, in a hypothetical scenario where the H1R knockout changes vascular dynamics in REM sleep, could this affect the results? 

5) Along the same lines, the authors report a surprisingly big increase in calcium signaling in REM sleep compared to NREM sleep - different from previous reports from other parts of the cortex. Could the astrocytic calcium signal be affected by hemodynamic artifacts, OR the correction performed using the 405 nm signal in such a way that it exaggerates the level of calcium signaling in REM sleep? If the 405/"isosbestic signal" is more affected by hemoglobin absorption than the 465 nm and 560 nm line, would not the compensation artificially elevate both the 465 and 560 line?

6) To the best of my knowledge, there are no other good methods to correct for hemodynamics artifacts than what the authors have done. However, a (supplementary) figure showing all the different signals (while stimulating with different LEDs) including transitions to and from REM, and the steps of signal processing performed would be a significant improvement. Moreover, to discuss this technical aspect in the discussion would improve the manuscript, and also raise the awareness of hemodynamic artifacts in sleep recordings (seems like many publications do not at all address this issue). 

Reviewer #2: In this manuscript by Taylor et al., the authors sought to determine if the wake-promoting neuromodulator histamine can act through astrocytes in the primary visual (V1) cortex to impact cortical arousal and sleep/wake behavior. These studies address important, fundamental questions regard the role of glia in sleep and wakefulness. Using a combination of ex vivo and in vivo preparations paired with dynamic imaging, electrophysiological, and genetic techniques, they report histamine differentially alters astrocyte intracellular calcium via the histamine H1 receptor (H1R) across sleep/wake states as well as adenosine activity specifically around rapid eye movement (REM) sleep. They also reported that conditional knockout (cKO) of astrocytic H1R in V1 had minimal effects on sleep/wake behavior but caused a small increase in time spent in wakefulness contrary to what would be predicted from this manipulation. The studies were generally well-designed, imaging aspects of these experiments were rigorous and technically well-executed, and the manuscript is clearly written with sound rationale. However, my primary concerns center around aspects of the execution of sleep/wake analyses and experimental design which could lead to inaccurate interpretations of the data. I detail these and other concerns below:

ESSENTIAL

1) The authors use a combination of multitaper spectral analysis and a Hidden Markov Model to automatically assign sleep/wake states to electroencephalographic (EEG) activity (i.e., score). It does not seem the electromyography (EMG) was used to support state detection except for in some REM sleep bouts. One major issue is that it does not appear that this autoscoring was manually/visually verified and corrected by an experimenter. Autoscoring approaches are notoriously known for inaccurately scoring REM sleep (i.e., distinguishing REM sleep from wakefulness) as well as state transitions even with additional measures like EMG. Given the data presented, there are a few indications that there are errors in state designation primarily with respect to REM sleep:

a. First, mice typically transition out of REM sleep directly to wakefulness (even if brief) before entering NREM sleep. Direct REM-to-NREM sleep transitions are not typically observed. However, the hypnograms shown in Figures 5B and 7B show few (if any) denoted REM episodes immediately followed by wakefulness. The authors also state that too few REM-to-wake transitions were observed for analysis, but this is inconsistent with mouse sleep physiology.

b. Second, we know from previous work in other cortical areas—and the authors state—that astrocyte intracellular calcium activity is highest during wakefulness and lower during REM sleep. In addition, the calcium activity during REM sleep is similar to NREM sleep, if not lower. However, the data in this manuscript are inconsistent with this previous work, which the authors acknowledge.

c. Third, previous work shows wake onset immediately after REM sleep is associated with large increases in astrocyte calcium activity that generally surpass that seen in NREM-to-wake transitions and sustained wakefulness. The data shown in Figure 5B depict the largest increases in astrocyte calcium are associated with REM sleep episodes/transitions. Based on previous work, I strongly suspect that these large calcium events instead occur during wakefulness rather than REM or NREM sleep as reported in this manuscript.

d. Related to the above point, I suspect the peaks shown in Figures 5G and 5H also likely reflect REM-to-wake transitions. For 5G, a peak occurs around 30 seconds after the denoted REM sleep onset. REM sleep episodes in mice last approximately 60 seconds on average. However, shorter REM episodes (e.g., 10 seconds and longer) are also common. It is not clear if the data shown in 5G and 5H only include REM episodes that are ≥ 60 s. But if shorter REM episodes are included, then these data likely reflect transitions to wakefulness (and perhaps subsequently to NREM sleep). REM-to-wake-to-NREM transitions are seen within a 60-s window in mice, and the pattern of astrocyte calcium activity in 5G seems like it could reflect such a transition pattern. I also suspect the peak just following the denoted NREM onset in 5H also reflects activity during wakefulness.

e. If EMG was not used to help distinguish between wakefulness and NREM, I also wonder if the ~30-s decrease in astrocyte calcium after NREM onset denoted in 5F might be a period of quiet wakefulness.

f. Finally, the authors state that REM sleep was sometimes determined without EMG but instead used the criteria that the bout had to follow a NREM bout and had to be ≤ 200 seconds. However, these criteria are equally applicable to wakefulness in mice. Therefore, this approach is prone to error. 

Given the inconsistencies between previous work and the current data, please compare the autoscoring against visual/manual scoring by an experienced experimenter to 1) validate (what seems to be) a relatively new approach for this team with respect to sleep analysis and 2) verify the accuracy of the currently reported state scorings. Without these data, it is difficult to accurately interpret the data presented in this manuscript.

2) Related to #1 and the multitaper spectral analysis, I wonder if a smaller window size would provide a more accurate starting point for autoscoring. From my understanding, multitaper spectral analysis has been used primarily on human EEG data. Because human EEG is more stable (i.e., fewer state transitions), we typically scored human EEG in 30-s epochs. This stability is in contrast to mice which have more frequent state transitions and are therefore scored using 4-s or 10-s epochs. Therefore, a window size of 4 s or 10 s might be more appropriate for this work to improve temporal resolution especially given that the EEG-determined states are being compared against continuous changes in calcium and adenosine.

3) It is unclear how long the duration of EEG/EMG recordings were for each experiment described in Figure 5, Figure 6, and Figure 7. The methods state 195 min recordings were captured under Fiber photometry and electrophysiology recordings. Was this true for experiments in Figure 6? Also, it seems that Figure 5 shows examples of ~250 min recordings.

4) Further detail regarding the process for transferring mice to recording cages and connecting mice to cables will be helpful. More specifically, how long were mice allowed to habituate to the recording environment and tether before each recording session? The Methods state mice habituated for at least one 2-h session before recordings began, but was this habituation period given before every recording? 

If not (and even if so), is it possible that H1R cKO mice need more time to habituate for Figure 6 experiments? If cKO mice needed longer to settle into the recording space, this could potentially account for the increased time spent in wakefulness. Therefore, increased time in wakefulness might be related to anxiety, for example, rather than sleep/wake regulation. Hypnograms shown in Figures 5B and 7B show that cKO mice spend more time in wakefulness at the beginning of the recording compared to WT. If the data are available, calculating onset to NREM sleep after being connected to the EEG/EMG tether/preamplifier could help answer that question.

5) The authors state that all experiments occurred during the light period in the Methods. I have a few concerns regarding the time-of-day during which the experiments were performed:

a. First, this is an important detail that should be mentioned in the Abstract, Results, and Discussion sections.

b. The authors should state more precisely when during the light period (e.g., Zeitgeber time) these experiments were performed. Time spent in states, distribution of states, EEG spectral power, and astrocyte calcium activity differ across the light period, so this can impact interpretation of the data if time-of-day is not carefully considered. If experiments were spread across the light period, was this timing at least balanced across conditions for each experiment?

6) Besides what's mentioned in #4 & 5, I wonder if time-of-day could account for the differences in time-in-state between Figure S5 and Figure 6. The fact that all mice are spending more time in wakefulness compared to NREM sleep in Figure 6B makes me wonder if these recordings started near light onset or at the end of the light period. This can be clarified by addressing #5. Or perhaps there was a disturbance in the room that prevented sleep. 

I also think that time-of-day and/or behavioral changes unrelated to sleep/wake regulation are more likely to account for these differences than EEG electrode placement. The authors speculate that placing the EEG electrode ipsilateral (rather than contralateral) to the AAV injection accounts for the differences with Figure S5 and reflects local sleep. However, a change in EEG placement is unlikely to result in a change in overall behavior. Placing EEG electrodes over different parts of the cortex can change the spectral qualities of the EEG, but I am skeptical that we'd see an EEG electrode over one part of the cortex show > 20% difference in vigilance state from an electrode over another part of the cortex in mice when EMG can be used to help confirm behavior/state. Therefore, EMG should be used to more accurately define sleep/wake states.

If there are EEG spectral differences that do not align with other behavioral measures like EMG, that is very interesting! But it would be best to discuss this state characterization in a more nuanced way than how we typically describe sleep/wake behavior as you'll have something more akin to a dissociated state. 

7) What was the rationale for conducting these studies during the light period? Because histamine is wake-promoting and we'd expect knocking out H1R to decrease wakefulness, it seems like this prediction would be best tested during the dark period when wake propensity is higher in mice. The increased astrocyte calcium activity during light-period wakefulness is interesting, but it might not generalize to the dark period since we often see manipulation affect sleep/wake parameters in one phase but not the other. It might be helpful to mention this in the Discussion.

MINOR

8) The authors speculate that calcium activity in cKO cells after histamine application might be due to being connected to the syncytium or neuronal H1R-driven responses. But is it also possible that histamine is activating other histamine receptors even if H1R is enriched on astrocytes?

9) Figure 4B: The text indicates that norepinephrine was added to slices after histamine application and that these applications were separated by at least 10 min (pg. 6, lines 33-34; pg. 7, lines 6-7, pg. 11, line 20). However, Figure 4B seems to indicate histamine was applied continuously and norepinephrine was added later, but during, histamine application. 

10) Please state how many slices from how many animals were used in Figure 4 and S4 experiments.

11) Please state how many animals were used in Figure 5 experiments.

12) How were sleep/wake bouts defined? Was a minimum duration threshold used (e.g., at least 30 s for wakefulness)? Please define this in the methods.

13) Pg. 8, lines 13-14: I wonder if the differences between in vivo and ex vivo calcium activity data might, at least in part, be due to different measurements used. In vivo data are described in terms of peak amplitude and peak frequency, but ex vivo data are described in terms of active pixels.

14) Pg. 8, line 11: I think 5F is supposed to be 5G.

15) Pg. 8, line 24: I think 5E is supposed to be 5F.

16) Pg. 8, line 25: I think Fig. 5E, G is supposed to be Fig. 5E, F.

17) Pg. 8, line 26: I think 5F is supposed to be 5G, H.

18) Figure S6C: should the green label be something different from Cre-GFP? Also, S6D seems to show Cre expression in the contralateral cortex.

19) When no Cre virus was injected (e.g., Figure 5), was anything added to supplement the volume so that WT and cKO were injected with the same volume? If not, please edit the Methods to reflect this detail.

20) Pg. 31, line 17: Is a ~5 mm diameter tip correct?

21) In the Methods, please state the vendor and product numbers for all of the AAVs. If they are custom, please state that. 

22) Pg. 35, lines 33-34: As stated, it reads like mice were anesthetized with buprenorphine and carprofen. 

23) Pg. 36: Please state the AAV titers used for the in vivo experiments. If they are the same as ex vivo experiments, please state that. 

Reviewer #3: This is an interesting study. Charlotte R. Taylor claimed that histamine increases astrocyte intracellular Ca2+ concentration via H1R in V1. Although V1 astrocyte-specific H1R deletion minimally affects sleep/wake architecture or cortical oscillations, they revealed a significant decrease in extracellular adenosine levels during REM sleep rather than wakefulness. However, there are major issues that need to be corrected before appropriate interpretation of the results can be done.

Major:

1) Previous studies suggest that histamine neural activity is high during wakefulness and low during sleep. However, is the actual extracellular histamine concentration in V1 also high during wakefulness and low during sleep? This is the key question of this study, so why not measure it using a GRAB sensor?

2) In the astrocyte H1R cKO experiments, I find it difficult to understand why the effects on sleep/wake alternation differ between Figure 5 and Figure 6. The authors concluded that "Thus, uni-hemispheric H1R deletion in cortical astrocytes could cause differences in time awake between hemispheres, consistent with local sleep differences observed across cortex," citing the study by Vyazovskiy, V. V. et al., as supporting evidence. However, their findings refer to differences on the scale of hundreds of milliseconds, which may influence cortical oscillations but not the sleep/wake state itself. Moreover, the cited study was conducted on rats, not mice, making its applicability less convincing.

3) The authors assessed the specificity of H1R cKO in astrocytes using immunohistochemistry. However, there is no information on the number of V1 astrocytes that undergo H1R deletion. Additionally, there are no reports on the variability in the H1R deletion among different animals. This is relevant to major comment #2), as I am concerned that the extent of H1R deletion may also influence sleep/wake effects in individual animals. Correlating the extent of H1R deletion with the sleep/wake outcomes would be valuable. Without this information, it is difficult to accurately evaluate the role of H1R in V1 astrocytes in sleep/wake regulation.

4) On page 7, the authors claimed that "However, neighboring WT astrocytes exhibited attenuated responses to NE after HA stimulation (Fig. 4B, D). While this may be due to exhaustion of Ca2+ stores since we saw no difference in WT and cKO Ca2+ responses to NE when slices were not previously stimulated with HA (Fig. 4C, F; Video 4), we think it is unlikely because NE responses in cerebellar astrocytes recover within 4 minutes after a single preceding NE stimulation15, and HA and NE addition were separated by at least 10 minutes here. Instead, our results could reflect HA-triggered changes in astrocyte activity that outlast acute H1R activation, consistent with the observation that the decrease in NE-triggered WT responses relative to neighboring cKO responses is not correlated with the time between HA and NE addition (Fig. 4E). Together, these data show that HA directly activates astrocytes via H1R, leading to acute Ca2+ elevations in H1R-expressing astrocytes and potential regulation of subsequent neuromodulator responses." To support this assumption, it is necessary to conduct an experiment of repetitive NE application on V1 astrocytes.

Minor:

1) In supplementary figure 1A, why is the increase in Ca2+ concentration occurring before time 0?

2) Are all astrocytes expressing H1R? Or is it differentially expressed in different brain regions? Please add this information.

3) In Figure 5, Isn't the REM offset simply due to the lower increase in Ca2+ during REM compared to WT? This could explain why the authors do not detect any change in the Wake-to-NREM transition. If the authors are unable to present REM-to-Wake results, they should be more cautious in their interpretation of the NREM offset.

4) When did the authors record EEG/EMG to validate sleep/wake state and cortical oscillations in ZT? Please mention this in the results and the methods sections.

5) In supplementary figure 6A, according to the AAV injection site, it appears to be V2 rather than V1.

---

## [Decision Letter · Decision Letter 2]

16 Jul 2025

Dear Dr Poskanzer,

Thank you for your patience while we considered your revised manuscript "Histamine-1-receptors regulate cortical astrocyte calcium and extracellular adenosine dynamics across sleep and wake" for publication as a Research Article at PLOS Biology. This revised version of your manuscript has been evaluated by the PLOS Biology editors, the Academic Editor and the original reviewers. 

As you will see in their comments, which are appended below, both reviewers 1 and 2 are fully satisfied by the revision. Reviewer 3 agrees that the study has been strengthened, but also raises two lingering concerns. Reviewer 3 suggests that you add an additional control study to strengthen the claims that 'H1R activity attenuates NE-triggered astrocytic Ca²⁺ elevation' and that you consider focusing the study on the ipsilateral EEG data only, or perhaps provide more insights into what accounts for sleep/wake differences between ipsi- and contralateral cohorts. Having discussed these points with the Academic Editor, we think that these last points can be addressed with textual changes. 

Specifically, while we appreciate that reviewer 3's request for an additional experiment using consecutive NE applications would strengthen the study, we do not think that generating this data is essential at this stage, as we think that the evidence provided is already strong enough to support the claim that astrocytic H1R modulates responses to NE. However, in the absence of additional data, we think it would be good to address this point by explicitly acknowledging, in the discussion, the alternative possibility raised by R3 that initial astrocyte activation, even independently of H1R, could influence subsequent calcium responses.

Regarding the contralateral EEG data - we would encourage you to keep in both the contralateral and ipsilateral data as we think that removing the contralateral data would result in a loss of valuable context regarding experimental variability. We think that Reviewer 3's comment could be addressed by providing further clarifications in the text around how stress responses or experimental setup differences could partly underlie the observed effects and by emphasizing that these comparisons, while informative, are not central to the primary conclusions.

Based on the reviews and our Academic Editor's assessment of your revision, we are likely to accept this manuscript for publication, provided you satisfactorily address the remaining points raised by reviewer 3 as detailed above. Please also make sure to address the following data and other policy-related requests.

**IMPORTANT: As you address the last reviewer comments, please also address the following editorial requests. 

1) TITLE: We would like to suggest a tweak to the title, to make clearer that the study uses astrocyte specific manipulations of H1R. If you agree, we suggest the title be changed to: 

"Cortical astrocyte histamine-1-receptors regulate calcium and extracellular adenosine dynamics across sleep and wake"

2) ETHICS STATEMENT: Please update the ethics statement in your methods statement to include the approval number for the animal care and use protocol approved by the UCSF IACUC. Please also include the specific national or international regulations/guidelines to which your animal care and use protocol adhered. Please note that institutional or accreditation organization guidelines (such as AAALAC) do not meet this requirement.

3) DATA: I see your data availability statement currently says "All data files are available from Dryad" - however I could not find a link to that data. Please update the data availability statement to provide a DOI and link to access the underlying data for your study, as I will need to check that this meets our reporting requirements. For the details of our data reporting policy, which requires that all data be made available without restriction, see here: (http://journals.plos.org/plosbiology/s/data-availability) 

Please also add a sentence to each figure legend pointing readers to this underlying data file. 

4) DATA NOT SHOWN: Please note that per journal policy, we do not allow the mention of "data not shown", "personal communication", "manuscript in preparation" or other references to data that is not publicly available or contained within this manuscript. I saw one instance of this in the figure legend for Supplementary Figure 3. Perhaps you can include that data in the figure?

5) CODE: Per journal policy, if you have generated any custom code during the course of this investigation, please make it available without restrictions. Please ensure that the code is sufficiently well documented and reusable, and that your Data Statement in the Editorial Manager submission system accurately describes where your code can be found. 

We expect to receive your revised manuscript within two weeks. 

*Published Peer Review History*

*Press*

Sincerely,

Luke

Lucas Smith, Ph.D.

Senior Editor

lsmith@plos.org

PLOS Biology

Reviewer remarks:

Reviewer #1: I have no further concerns/comments. I commend the authors for critically re-evaluating some of their main findings and transparently reporting the new results.

Reviewer #2: The authors carefully and thoughtfully addressed all my previous concerns, and I have no further comments. Congratulations to the authors!

Reviewer #3, Tomomi Tsunematsu (note, reviewer 3 has signed this review): Here, the authors present a revised manuscript that is much improved over the previous version. Notably, the improvement in the accuracy of sleep/wakefulness state classification is a valuable strength of the study. Nevertheless, there are still several issues of concern that require classification and revision.

Major

1) The authors demonstrated that, following administration of a low concentration of HA that is unlikely to deplete Ca²⁺ stores, subsequent NE application induced a smaller astrocytic response in WT compared to H1R cKO astrocytes. Furthermore, a single NE application, H1R cKO astrocytes, exhibited a greater response than WT astrocytes. However, based on these experiments alone, it is insufficient to conclude that "H1R activity attenuates NE-triggered astrocytic Ca²⁺ elevation." The possibility remains that astrocyte activation itself—specifically, the first increase in intracellular Ca²⁺ concentration—regardless of whether it is mediated by H1R or not, alters the subsequent response. Although I agree with the authors' argument that Ca²⁺ store depletion is unlikely under these conditions, definitive evidence that this effect is H1R-dependent requires additional experiments. In particular, two consecutive NE applications would be necessary. If the response to the second NE application is comparable to the first response in both H1R cKO and WT astrocytes, it would support the notion that H1R mediates this attenuation. Alternatively, if both groups show a reduced response to the second NE application relative to the first, it would indicate that astrocytes inherently exhibit reduced responsiveness upon repeated activation, independent of H1R signaling.

2) The authors discuss sleep/wakefulness state based on both contralateral and ipsilateral EEG recordings, as well as the potential influence of fiber implantation. However, it is important to note that both H1R cKO and WT mice were equally subjected to the potential effects of the implanted fiber. While I agree that fiber implantation could slightly alter sleep/wakefulness architecture, the observed differences between cKO and WT mice require further explanation. For instance, it is possible that cKO mice are more vulnerable to stress. To support this hypothesis, additional data demonstrating differences in stress tolerance—such as the impact of sleep deprivation—would be necessary. That said, I am not convinced that this line of discussion is central to the main conclusions of the present study. Given that these results could potentially confuse readers and are not essential to the primary message of the manuscript, I would suggest that the authors consider presenting only the ipsilateral EEG data and focusing the discussion accordingly.

Minor

1) Please reverse the order of the graphs for WT and H1R cKO mice in Figure 7C.

---

## [Editor Report · Decision Letter 3]

20 Aug 2025

Dear Kira,

Thank you for the submission of your revised Research Article "Cortical astrocyte histamine-1-receptors regulate intracellular calcium and extracellular adenosine dynamics across sleep and wake" for publication in PLOS Biology and for addressing the last reviewer and editorial requests in this revision. On behalf of my colleagues and the Academic Editor, Cagla Eroglu, I am pleased to say that we can in principle accept your manuscript for publication, provided you address any remaining formatting and reporting issues. These will be detailed in an email you should receive within 2-3 business days from our colleagues in the journal operations team; no action is required from you until then. Please note that we will not be able to formally accept your manuscript and schedule it for publication until you have completed any requested changes.

As one additional note - I saw that you provided an updated 'Data Availability Statement' in the file inventory for your submission. I have taken the liberty of moving this to the relevant section of our editorial manager system, as that is the version that will be published. You can delete the data availability statement file from your submission. 

The 'Data Availability Statement' now reads: "All data underlying the results are available in Dryad (DOI: 10.5061/dryad.2280gb64x). The analysis scripts and notebooks used to generate the figures are archived in Zenodo (DOI: 10.5281/zenodo.16809849). Both resources are publicly available without restriction."

PRESS

We frequently collaborate with press offices. If your institution or institutions have a press office, please notify them about your upcoming paper at this point, to enable them to help maximize its impact. If the press office is planning to promote your findings, we would be grateful if they could coordinate with biologypress@plos.org. If you have previously opted in to the early version process, we ask that you notify us immediately of any press plans so that we may opt out on your behalf.

Sincerely, 

Luke

Lucas Smith, Ph.D.

Senior Editor

PLOS Biology

lsmith@plos.org